# Time-Marching Quantum Algorithm for Simulation of Nonlinear Lorenz Dynamics

**DOI:** 10.3390/e27080871

**Published:** 2025-08-17

**Authors:** Efstratios Koukoutsis, George Vahala, Min Soe, Kyriakos Hizanidis, Linda Vahala, Abhay K. Ram

**Affiliations:** 1School of Electrical and Computer Engineering, National Technical University of Athens, 15780 Zographou, Greece; kyriakos@central.ntua.gr; 2Department of Physics, William & Mary, Williamsburg, VA 23187, USA; 3Department of Mathematics and Physical Sciences, Rogers State University, Claremore, OK 74017, USA; msoe.rsu@gmail.com; 4Department of Electrical and Computer Engineering, Old Dominion University, Norfolk, VA 23529, USA; lvahala@odu.edu; 5Plasma Science and Fusion Center, Massachusetts Institute of Technology, Cambridge, MA 02139, USA; abhay@psfc.mit.edu

**Keywords:** time-marching quantum algorithm, recursive structure, Hadamard product, SVD block encoding, linear combination of unitaries, nonlinear ordinary differential equations, Lorenz system

## Abstract

Simulating nonlinear classical dynamics on a quantum computer is an inherently challenging task due to the linear operator formulation of quantum mechanics. In this work, we provide a systematic approach to alleviate this difficulty by developing an explicit quantum algorithm that implements the time evolution of a second-order time-discretized version of the Lorenz model. The Lorenz model is a celebrated system of nonlinear ordinary differential equations that has been extensively studied in the contexts of climate science, fluid dynamics, and chaos theory. Our algorithm possesses a recursive structure and requires only a linear number of copies of the initial state with respect to the number of integration time-steps. This provides a significant improvement over previous approaches, while preserving the characteristic quantum speed-up in terms of the dimensionality of the underlying differential equations system, which similar time-marching quantum algorithms have previously demonstrated. Notably, by classically implementing the proposed algorithm, we showcase that it accurately captures the structural characteristics of the Lorenz system, reproducing both regular attractors–limit cycles–and the chaotic attractor within the chosen parameter regime.

## 1. Introduction

Quantum computing provides a paradigm shift in how we perceive computation by offering the prospect of accelerating certain computational tasks beyond the capabilities of classical computers. In this direction, there is much interest in seeing how quantum computing and information science can be employed to solve important problems in classical physics, and in particular, whether quantum computers can be harnessed to speed-up the simulation of complex, nonlinear dissipative classical systems exhibiting classical turbulence and chaos [1]. Naturally, these systems pose significant challenges for integration into the linear and unitary quantum algorithmic framework, as their dynamics are non-norm preserving and nonlinear.

Although the quantum algorithmic study of dissipative nonlinear classical systems is still in its infancy, various techniques have been employed to adapt the underlying nonlinear differential equations and corresponding dynamics to the framework of quantum simulation. These include Carleman linerization [2,3,4,5], the Koopman–von Neumann (KvN) formulation of classical mechanics [6,7], second quantization methods [8,9,10], measurement-based approaches [11], quantum variational algorithms [12,13,14], and quantum time-marching schemes [15,16,17,18,19]. While each method offers certain advantages, the first three are impeded, both in terms of quantum algorithmic advantage and correct physics, when applied to chaotic systems [11,20,21,22].

Carleman and KvN-based techniques involve an analytic expansion of the solution to a partial differential equation in terms of an order parameter, thereby embedding the finite-dimensional nonlinear system into an infinite-dimensional linear one [23]. By truncating the resulting hierarchy of linear equations, quantum simulation algorithms can then be applied to the finite-dimensional linear approximation. However, intuitively speaking, the effectiveness of such linear embedding techniques depends critically on the presence of the Painlevé property [24], which ensures the convergence of the analytic expansion. This typically restricts the successful application of linear embeddings to regular dynamics [20], such as the one-dimensional Burgers equation [2] and the advection–diffusion equation [7], both known to possess the Painlevé property. Another issue with these linear embedding techniques is that, while they enable the quantum simulation of nonlinear classical systems in specific regimes, the necessary projection into a finite-dimensional space can introduce numerical artifacts that lead to inaccurate results [22]. Finally, measurement-based and second quantization techniques are dominated by quantum information scrambling for long simulation times [9,11,25].

Meanwhile, quantum implementations of time-marching schemes have shown promising applications in solving nonlinear differential equations, such as a 1D cubic nonlinear ordinary differential equation [18], as well as Burgers equation for high speed flows [19]. In this paper, we consider the latter approaches by proposing a quantum implementation of classical explicit and high-order time-advancement schemes, aiming to develop a quantum solver capable of accurately capturing the chaotic behavior of the Lorenz system. Despite its simplicity, the Lorenz system remains one of the most extensively studied chaotic systems, exhibiting all the essential features of non-Hamiltonian chaos. It has found widespread applications in fluid dynamics, weather forecasting, and chaos theory, among other areas.

To implement the time-advancement of a system of nonlinear differential equations, we propose a two-stage approach similar to the method presented in [15]. Specifically, by selecting a temporal discretization scheme, the update of the state vector is expressed as a linear operator acting on an augmented nonlinear state in closed form. The single-time-step evolution employs the Hadamard product to prepare the nonlinear state, followed by a block encoded version of the linear time-update operator. A key feature of the proposed quantum solver is the adoption of a quantum re-usage of states [19], resulting in a recursive structured quantum time-marching algorithm that maintains the previously established quantum speed-up while requiring only a linear number of state copies with respect to the total number of integration steps $N_{t}$. We demonstrate the effectiveness of our construction for a second-order discretized version of the Lorenz system by reproducing the correct qualitative behavior of both chaotic and regular attractors.

The structure of the paper is as follows: Section 2 introduces the basic equations describing the Lorenz dynamics and the respective first- and second-order discretization schemes. The two-stage single-time-step evolution is presented at the beginning of Section 3, consisting of a nonlinear state preparation followed by a non-unitary linear evolution. To create the nonlinear states, we build on the Hadamard product in Section 3.1.1 and Section 3.1.2, while the block encoding technique for implementing the non-unitary evolution operator is presented in Section 3.2. In Section 3.3 the complete quantum time-marching algorithm is illustrated. Then, in Section 3.4, the complexity of the quantum algorithm, both in terms of quantum resource scaling and the number of oracle queries, is analyzed. Its recursive and post-selective features are also discussed. A comparison of the classically implemented results with those generated by an adaptive, high-order Mathematica solver is given in Section 4. Finally, in Section 5, we summarize our findings and outline directions for future research.

## 2. The Lorenz Model

The Lorenz model [26] was first explored in the study of heat conduction in atmospheric physics. It consists of a set of 3 ordinary differential equations with two quadratic nonlinear terms and 3 positive free parameters σ,ρ,β,(1)$$\frac{dx}{dt}=\sigma (y-x),$$(2)$$\frac{dy}{dt}=x(\rho -z)-y,$$(3)$$\frac{dz}{dt}=xy-\beta z.$$
The constants σ and ρ are the system’s parameters proportional to the Prandtl number and Rayleigh number, and β is the ratio of width to height of the fluid layer. The Lorenz Equations (Equation 1)–(Equation 3) are dissipative since the volume V(t) in phase space enclosed by a surface S(t) evolves as(4)$$\frac{dV(t)}{dt}=\int_{S(t)}F\cdot d^{2}a=\int_{V(t)}\nabla \cdot Fd^{3}v=-\left(\sigma +1+\beta \right)V\left(t\right),$$
where the vector field F has components (dx/dt,dy/dt,dz/dt) and $d^{2}a,\;\;d^{3}v$ are the corresponding differential surface and volume elements. Thus, there is an exponential contraction of the three dimensional phase space volume to a set of measure zero(5)$$V\left(t\right)=e^{-(\sigma +1+\beta )t}V\left(0\right)\to 0\;\mathrm{as}\;t\to \infty .$$
Varying the free parameters in the system leads to an incredibly rich bifurcation phenomena, from periodic limit cycles to chaotic attractors with dimension less than 3.

### Selection of the Time-Marching Scheme

Applying a first-order forward Euler finite difference scheme to Equations (Equation 1)–(Equation 3) in the temporal domain [t,t+T] we obtain in vector form,(6)$$x_{n+1}=x_{n}+\delta tf\left(x_{n}\right),\;x_{n}=\left[\begin{array}{c} x_{n}\\ y_{n}\\ z_{n} \end{array}\right],$$
where $x_{n}\equiv x\left(t+n\delta t\right),\;\;n=1,2,\ldots ,N_{t},\;\;N_{t}=T/\delta t$ and f is a polynomial vector function such that $f\left(x\right)={\hat{C}}_{1}x+{\hat{C}}_{2}x^{⊗2}$ with ${\hat{C}}_{1,2}$ being the coefficient matrices. By embedding the nonlinear terms of Equation (Equation 6) into the state vector $\psi_{n}^{nl}$ a closed set of linear recursive equations is obtained,(7)$$\psi_{n+1}={\hat{A}}_{1}\psi_{n}^{nl},$$
with(8)$$\psi_{n+1}=\left[\begin{array}{c} x_{n+1}\\ 0_{5\times 1} \end{array}\right],\;\psi_{n}^{nl}=\left[\begin{array}{c} x_{n}\\ x_{n}y_{n}\\ x_{n}z_{n}\\ 0_{3\times 1} \end{array}\right],$$
and(9)$${\hat{A}}_{1}=\left[\begin{array}{cc} {\hat{A}}_{1} & 0_{3\times 3}\\ 0_{5\times 3} & 0_{5\times 5} \end{array}\right],\;{\hat{A}}_{1}=\left[\begin{array}{ccccc} 1-\sigma \delta t & \sigma \delta t & 0 & 0 & 0\\ \rho \delta t & 1-\delta t & 0 & 0 & -\delta t\\ 0 & 0 & 1-\beta \delta t & \delta t & 0 \end{array}\right].$$

However, the first-order Euler scheme runs into numerical problems for stiff differential equations, which are not eliminated even for δt<<1 [14,27]. These computational inaccuracies lead to an inability to reproduce the correct Lorenz dynamics. For instance, the first-order Euler method for the parameters σ=10,ρ=28,β=0.55 does not reproduce the period-2 limit cycle as found by standard fourth-order Runge–Kutta solvers. Since we seek an accurate quantum algorithm for the nonlinear dissipative Lorenz system, we consider a 2nd order two-stage Runge–Kutta extension of the simple Euler scheme.

The employed second-order difference scheme is a predictor-corrector of the form [28],(10)$$\begin{array}{cc} {\tilde{x}}_{n+1} & =x_{n}+\delta tf\left(x_{n}\right), \end{array}$$(11)$$\begin{array}{cc} x_{n+1} & =x_{n}+\frac{\delta t}{2}\left[f\left(x_{n}\right)+f\left({\tilde{x}}_{n+1}\right)\right]. \end{array}$$
Notice that the second-order time marching scheme in Equations (Equation 10) and (Equation 11) is explicit and can be written in an analogous form to that of Equation (Equation 7),(12)$$\psi_{n+1}={\hat{A}}_{2}\psi_{n}^{nl},$$
where now(13)$$\psi_{n+1}=\left[\begin{array}{c} x_{n+1}\\ 0_{13\times 1} \end{array}\right],\;\psi_{n}^{nl}=\left[\begin{array}{c} x_{n}\\ x_{n}y_{n}\\ x_{n}z_{n}\\ y_{n}z_{n}\\ x_{n}y_{n}^{2}\\ x_{n}^{2}y_{n}\\ x_{n}^{2}z_{n}\\ x_{n}y_{n}z_{n}\\ 0_{6\times 1} \end{array}\right],$$
and(14)$${\hat{A}}_{2}=\left[\begin{array}{cc} {\hat{A}}_{2} & 0_{3\times 6}\\ 0_{13\times 10} & 0_{13\times 6} \end{array}\right],\;{\hat{A}}_{2}=\left[\begin{array}{cccccccccc} a_{1} & a_{2} & 0 & 0 & a_{5} & 0 & 0 & 0 & 0 & 0\\ b_{1} & b_{2} & 0 & 0 & b_{5} & b_{6} & b_{7} & b_{8} & 0 & 0\\ 0 & 0 & c_{3} & c_{4} & 0 & 0 & 0 & 0 & c_{9} & c_{10} \end{array}\right].$$
The elements of matrix ${\hat{A}}_{2}$ in Equation (Equation 14) read,(15)$$\begin{array}{cc} \; & a_{1}=1-\sigma \delta t+\frac{\sigma \delta t^{2}}{2}\left(\rho +\sigma \right),\;a_{2}=\sigma \delta t\left(1-\frac{\delta t}{2}-\frac{\sigma \delta t}{2}\right),\;a_{5}=-\frac{\sigma \delta t^{2}}{2}, \end{array}$$(16)$$\begin{array}{cc} \; & b_{1}=\rho \delta t\left(1-\frac{\delta t}{2}-\frac{\sigma \delta t}{2}\right),\;b_{2}=1-\delta t+\frac{\delta t^{2}}{2}+\frac{\rho \sigma \delta t^{2}}{2},\;b_{5}=\frac{\delta t}{2}\left[-1+\delta t-(1-\beta \delta t)(1-\sigma \delta t)\right], \end{array}$$(17)$$\begin{array}{cc} \; & b_{6}=-\frac{\sigma \delta t^{2}\left(1-\beta \delta t\right)}{2},\;b_{7}=-\frac{\sigma \delta t^{3}}{2},\;b_{8}=-\frac{\delta t^{2}\left(1-\sigma \delta t\right)}{2}, \end{array}$$(18)$$\begin{array}{cc} \; & c_{3}=1-\beta \delta t+\frac{\beta^{2}\delta t^{2}}{2},\;c_{4}=\frac{\delta t}{2}\left[1-\beta \delta t+\left(1-\delta t\right)\left(1-\sigma \delta t\right)+\sigma \rho \delta t^{2}\right], \end{array}$$(19)$$\begin{array}{cc} \; & c_{9}=-\frac{\delta t^{2}}{2}\left(1-\sigma \delta t\right),\;c_{10}=-\frac{\sigma \delta t^{2}}{2}. \end{array}$$
As expected, these coefficients agree with the Euler 1st order coefficients in Equation (Equation 9), up to terms of $O(\delta t^{2})$.

In the next section, the building blocks of the quantum algorithm for implementing the second-order scheme in Equation (Equation 12) are presented.

## 3. The Quantum Algorithm

Based on Equations (Equation 7) or (Equation 12), the single-time-step evolution $\psi_{n}\to \psi_{n+1}$ involves the operations outlined in Table 1.

From a classical computing perspective, the single-time-step evolution shown in Table 1 is explicit and straightforward; however, its implementation on a quantum computer requires careful treatment. This is because steps 2 and 3 are not compatible with the linear and unitary framework of quantum computing. From this point onward we adopt the Dirac bra–ket notation |ψ〉 to indicate the normalized, amplitude encoded, quantum variant of the classical state ψ. For instance, the discretized classical state $\psi_{n}$ from Equation (Equation 8) is amplitude encoded as a 3-qubit normalized quantum state $|\psi_{n}\rangle$: (20)$$|\psi_{n}\rangle =\frac{x_{n}|0\rangle +y_{n}|1\rangle +z_{n}|2\rangle }{\sqrt{x_{n}^{2}+y_{n}^{2}+z_{n}^{2}}},$$
with the binary representation of the basis states |*i*〉 given by $|0\rangle ={|000\rangle }_{b},\;\;|1\rangle ={|001\rangle }_{b},\;\;|2\rangle ={|010\rangle }_{b}$, etc. Each classical state ψ corresponds to a unique amplitude encoded quantum state |ψ〉. However, the reverse is not true: a |ψ〉 state can represent a whole family of classical states of the form rψ with r>0. Nevertheless, ψ is a solution of the Lorenz system, whereas rψ for r≠1 is not. Thus, this ambiguity is resolved by choosing r=1. Since the linear matrix $\hat{A}$ is non-unitary, $\langle \psi |{\hat{A}}^{†}\hat{A}|\psi \rangle \neq \langle \psi |\psi \rangle$, and the mapping $|\psi_{n}\rangle \to |\psi_{n}^{nl}\rangle$ is nonlinear it is necessary to express these operations within the admissible linear and unitary quantum framework.

In addition to these challenges, extending the single-time-step procedure to consecutive applications on a quantum computer is not straightforward. This stems from the fact that to prepare the nonlinear state $|\psi^{nl}\rangle$, one must create multiple copies of the initial state $|\psi_{n}\rangle$, i.e., this initial state can be quantum prepared and is known.

We address these issues explicitly by presenting the quantum blocks for a single-step evolution and then building them into a complete quantum algorithmic multi-step evolution.

### 3.1. Preparation of the Nonlinear States

The corresponding nonlinear states $\psi_{n}^{nl}$ for the first- and second-order discretized Lorenz system in Equations (Equation 8) and (Equation 13), exhibit polynomial nonlinearities with respect to the amplitudes $x_{n},y_{n},z_{n}$ of the $\psi_{n}$ state. Therefore, the first step is to determine how to implement on a quantum computer the following nonlinear transformation of the amplitudes $\psi_{i}$, for an *n*-qubit quantum state $|\psi \rangle =\sum_{i=0}^{2^{n}-1}\psi_{i}|i\rangle$,(21)$$|\psi \rangle \to \sum_{i}P_{i}\left(\psi_{i}\right)|i\rangle ,\;P_{i}\left(\psi_{i}\right)=\sum_{j=1}^{K}a_{j}\psi_{i}^{j},$$
where $a_{j}$ are the coefficients and *K* the degree of the polynomials $P_{i}\left(\psi_{i}\right)$.

In the following section we describe the implementation procedure of the nonlinear transformation in Equation (Equation 21), by using the Hadamard product. Then, we showcase how this approach can serve as a foundation for constructing arbitrary polynomially nonlinear states, and in particular, those arising in the Lorenz system.

#### 3.1.1. The Hadamard Product

The Hadamard product of two *n* and *m* qubit states |ψ〉 and |ϕ〉 respectively, is defined as follows(22)$$|\psi \rangle ⊙|\phi \rangle \equiv |\psi ⊙\phi \rangle =\sum_{i=0}^{2^{k}-1}\psi_{i}\phi_{i}|i\rangle ,\;k=max\left\{n,m\right\}.$$
Boldface quantum states denote multi-qubit states, while non-boldface bra–kets refer to the computational basis. Quantum implementation of the Hadamard product (Equation 22) for the case of single-qubit states was first presented in [29] and is illustrated in Figure 1.

As depicted in Figure 1, successful implementation of the Hadamard product is conditioned on the measurement of the 0-bit in the first register. Hereafter, the normalization factor in the successful post-measurement outcome will be omitted. For |ψ〉=|ϕ〉, we obtain $|\psi ⊙\psi \rangle =\psi_{0}^{2}|0\rangle +\psi_{1}^{2}|1\rangle$ after tracing out the ancillary |0〉 state.

Generalizing the Hadamard product implementation to two *n*-qubit states requires—instead of just a single Controlled-NOT (CNOT) gate—the introduction of a multi-control variant ${\hat{U}}_{select}^{h}$ through powers of the shift operator ${\hat{S}}_{-}$,(23)$${\hat{U}}_{select}^{h}=\sum_{k=0}^{2^{n}-1}{\hat{S}}_{-}^{k}⊗|k\rangle \langle k|,\;{\hat{S}}_{-}^{0}=\hat{1},\;{\hat{S}}_{-}|k\rangle =|k-1\rangle .$$
Indeed, with operator ${\hat{U}}_{select}^{h}$ acting on the 2n-qubit composite state |ψ〉⊗|ϕ〉 one obtains(24)$$\begin{array}{cc} {\hat{U}}_{select}^{h}\left(|\psi \rangle ⊗|\phi \rangle \right) & =\sum_{ijk}\psi_{i}\phi_{j}{\hat{S}}_{-}^{k}|i\rangle \langle k|j\rangle |k\rangle =\sum_{ij}\psi_{i}\phi_{j}|i-j\rangle |j\rangle \end{array}$$(25)$$\begin{array}{cc} \; & =\sum_{i=j}\psi_{i}\phi_{i}|0\rangle |i\rangle +\sum_{i\neq j}\psi_{i}\phi_{j}\underset{\neq |0\rangle }{\underbrace{|i-j\rangle }}|j\rangle . \end{array}$$
From Equation (Equation 25), a successful measurement in the first register with respect to the 0-bit provides the Hadamard product transformation between the two *n*-qubit states. Figure 2 presents the schematic quantum circuit implementations of the |ψ⊙ψ〉 transformation in terms of powers of the shift operator ${\hat{S}}_{-}^{k}$ along with the corresponding circuit implementation for the ${\hat{S}}_{-}$.

Decomposition of the ${\hat{S}}_{-}$ operator (Figure 2) into CNOTs and single-qubit gates scales as $O(n^{2})$ [32]. Therefore, the implementation cost of the full sequence of ${\hat{S}}^{k}$ gates in Figure 2 scales as $O(n^{2}2^{n+1})$. Given that the *n*-qubit uniformly controlled operations can be implemented within $O(2^{n})$ elementary gates [33], the overall implementation cost for the ${\hat{U}}_{select}^{h}$ which realizes the *n*-qubit Hadamard product is $O[2^{n}\left(2n^{2}+1\right)]$.

Extension of the Hadamard product to an *N*-tuple of the *n*-qubit state |ψ〉,(26)$$|\underset{N\;\;\mathrm{times}}{\underbrace{\psi ⊙\psi ⊙\ldots ⊙\psi }}\rangle \equiv |{⊙}^{N}\psi \rangle =\sum_{i=0}^{2^{n}-1}\psi_{i}^{N}|i\rangle ,\;N\geq 2,$$
can be realized through *N* copies of the |ψ〉 acting with the ${\hat{U}}_{h}^{N}$ operator as in Equation (Equation 23),(27)$${\hat{U}}_{h}^{N}=\sum_{k=0}^{2^{n}-1}\underset{N-1\;\;\mathrm{times}}{\underbrace{{\hat{S}}_{-}^{k}⊗\ldots ⊗{\hat{S}}_{-}^{k}}}⊗|k\rangle \langle k|.$$
Notice that for N=1, ${\hat{U}}_{h}={\hat{1}}_{2^{n}\times 2^{n}}$ and for N=2, ${\hat{U}}_{h}^{2}={\hat{U}}_{select}^{h}$ from Equation (Equation 22). Based on the previous discussion, operator ${\hat{U}}_{h}^{N}$ can be implemented within $O[2^{n}\left(2n^{2}N+1\right)]$ single-qubit and CNOT gates.

Finally, we adopt the following notation(28)$${\hat{U}}_{h}^{k}|\psi^{⊗l}\rangle \equiv |{⊙}^{k}\psi \rangle ,\;k\leq l,$$
indicating that the ${\hat{U}}_{h}^{k}$ operator acts locally on the first *k* copies from the *l*-fold tensor product of the |ψ〉 states, with rest of the l−k states omitted to keep the notation uncluttered.

#### 3.1.2. Implementing the Lorenz Nonlinear States

Having established the essential elements for the unitary representation of the Hadamard product, the nonlinear transformation in Equation (Equation 21) can be implemented by employing the Hadamard product in conjunction with the Linear Combination of Unitaries (LCU) method [34]. Specifically, using the definition in Equation (Equation 27), the nonlinear polynomial transformation in Equation (Equation 21) can be expressed as a weighted sum of Hadamard product ${\hat{U}}_{h}^{j}$ gates [35],(29)$${\hat{U}}_{nl}=\sum_{j=1}^{K}a_{j}{\hat{U}}_{h}^{j},\;a_{j}>0.$$
Acting with ${\hat{U}}_{nl}$ on the state $|\psi^{⊗K}\rangle$ and using the definitions in Equations (Equation 26) and (Equation 28) we obtain(30)$${\hat{U}}_{nl}|\psi^{⊗K}\rangle =\sum_{j=1}^{K}a_{j}|{⊙}^{j}\psi \rangle =\sum_{j=1}^{K}\sum_{i=0}^{2^{n}-1}a_{j}\psi_{i}^{j}|i\rangle ,$$
which coincides with the transformation of Equation (Equation 21).

Following the LCU implementation lemma [36], we define the following unitary operators,(31)$$\begin{array}{cc} \; & {\hat{U}}_{prep}^{LCU}|0^{⊗m}\rangle =\frac{1}{\sqrt{a}}\sum_{j=1}^{K}\sqrt{a_{j}}|j\rangle ,\;m=\log_{2}K,\;a=\sum_{j=1}^{K}a_{j}, \end{array}$$(32)$$\begin{array}{cc} \; & {\hat{U}}_{select}^{LCU}=\sum_{j=1}^{K}|j\rangle \langle j|⊗{\hat{U}}_{h}^{j}, \end{array}$$
which facilitate the circuit implementation of operator ${\hat{U}}_{nl}$ from Equation (Equation 29), as illustrated in Figure 3.

Introducing into Equation (Equation 29) the unitary multi-CNOT gates $\hat{G}$ that appropriately alternate the amplitudes $\psi_{i}$ within the state |ψ〉, we can prepare any nonlinear polynomial state. For example, suppose we wish to implement the following nonlinear 2-qubit state,(33)$$|\psi^{nl}\rangle =|\psi \rangle ⊗|\psi \rangle =\left[\begin{array}{c} \psi_{0}^{2}\\ \psi_{0}\psi_{1}\\ \psi_{1}\psi_{0}\\ \psi_{1}^{2} \end{array}\right].$$
This type of term arises in the first-order difference scheme, Equation (Equation 8). Then, the nonlinear state (Equation 33) can be written,(34)$$|\psi^{nl}\rangle ={\hat{G}}_{2}|\phi \rangle ⊙{\hat{G}}_{2}|\phi \rangle +{\hat{G}}_{1}|\phi \rangle ⊙{\hat{G}}_{0}|\phi \rangle ,\;|\phi \rangle =|0\rangle |\psi \rangle ,$$
where the operators $\hat{G}$ are(35)$${\hat{G}}_{0}=\left[\begin{array}{cccc} 0 & 0 & 0 & 1\\ 1 & 0 & 0 & 0\\ 0 & 1 & 0 & 0\\ 0 & 0 & 1 & 0 \end{array}\right],\;{\hat{G}}_{1}=\left[\begin{array}{cccc} 0 & 0 & 1 & 0\\ 0 & 1 & 0 & 0\\ 1 & 0 & 0 & 0\\ 0 & 0 & 0 & 1 \end{array}\right],\;{\hat{G}}_{2}=\left[\begin{array}{cccc} 1 & 0 & 0 & 0\\ 0 & 0 & 0 & 1\\ 0 & 0 & 1 & 0\\ 0 & 1 & 0 & 0 \end{array}\right].$$
Therefore, the implementation operator ${\hat{U}}_{nl}$ is given by(36)$${\hat{U}}_{nl}={\hat{U}}_{h}^{2}{\hat{G}}_{2}^{⊗2}+{\hat{U}}_{h}^{2}\left({\hat{G}}_{1}⊗{\hat{G}}_{0}\right).$$

Quantum circuit implementation of ${\hat{U}}_{nl}$ in Equation (Equation 36) follows the same application of the LCU method as in Figure 3 and is explicitly depicted in Figure 4.

The $\hat{G}$ gates alternating the amplitudes can be explicitly implemented through the Gray code [37]. In particular, ${\hat{G}}_{0}={\hat{1}}_{2\times 2}⊗|1\rangle \langle 0|+\hat{X}⊗|0\rangle \langle 1|$, ${\hat{G}}_{1}={\hat{1}}_{2\times 2}⊗|1\rangle \langle 1|+\hat{X}⊗|0\rangle \langle 0|$ and ${\hat{G}}_{2}={\hat{1}}_{2\times 2}⊗|0\rangle \langle 0|+\hat{X}⊗|1\rangle \langle 1|$.

Finally, the full nonlinear state $|\psi_{n}^{nl}\rangle$ for the first-order discretized Lorenz model in Equation (Equation 8) is determined though the operator,(37)$${\hat{U}}_{nl}={\hat{U}}_{h}^{1}+{\hat{U}}_{h}^{2}\left({\hat{G}}_{1}⊗{\hat{G}}_{0}\right),\;{\hat{U}}_{h}^{1}={\hat{1}}_{8\times 8},$$
with(38)$${\hat{G}}_{0}=\left[\begin{array}{cccccccc} 0 & 0 & 0 & 1 & 0 & 0 & 0 & 0\\ 0 & 0 & 0 & 0 & 1 & 0 & 0 & 0\\ 0 & 0 & 0 & 0 & 0 & 1 & 0 & 0\\ 1 & 0 & 0 & 0 & 0 & 0 & 0 & 0\\ 0 & 1 & 0 & 0 & 0 & 0 & 0 & 0\\ 0 & 0 & 1 & 0 & 0 & 0 & 0 & 0\\ 0 & 0 & 0 & 0 & 0 & 0 & 1 & 0\\ 0 & 0 & 0 & 0 & 0 & 0 & 0 & 1 \end{array}\right],\;{\hat{G}}_{1}=\left[\begin{array}{cccccccc} 0 & 0 & 0 & 1 & 0 & 0 & 0 & 0\\ 0 & 0 & 0 & 0 & 1 & 0 & 0 & 0\\ 0 & 0 & 0 & 0 & 0 & 1 & 0 & 0\\ 0 & 1 & 0 & 0 & 0 & 0 & 0 & 0\\ 1 & 0 & 0 & 0 & 0 & 0 & 0 & 0\\ 0 & 0 & 0 & 0 & 0 & 0 & 1 & 0\\ 0 & 0 & 1 & 0 & 0 & 0 & 0 & 0\\ 0 & 0 & 0 & 0 & 0 & 0 & 0 & 1 \end{array}\right].$$
Therefore, to implement the nonlinear state $|\psi_{n}^{nl}\rangle$ for the second-order Lorenz discretization in Equation (Equation 13), one simply replaces the two 2-qubit copies in Figure 4 with three 4-qubit copies of state $|\psi_{n}\rangle$.

### 3.2. Block Encoding of Non-Unitary Matrix $\hat{A}$

Implementing a non-unitary $2^{n}\times 2^{n}$ matrix $\hat{A}$ requires a (m,a)-block encoding of the form,(39)$${\hat{U}}_{A}=\overset{2^{m}\mathrm{matrix}\;\mathrm{elements}}{\overbrace{\left[\begin{array}{cccc} \hat{A}/a & * & \ldots & *\\ {}* & * & \ldots & *\\ \vdots & \vdots & \ddots & \vdots \\ {}* & * & \ldots & * \end{array}\right]}},$$
which corresponds to a dilation of the $2^{n}\times 2^{n}$ matrix $\hat{A}$ into a larger, $2^{n+m}\times 2^{n+m}$ unitary matrix ${\hat{U}}_{A}$ with $||\hat{A}/a||\leq 1$. The * elements in Equation (Equation 39) represent proper sub-matrices so that the ${\hat{U}}_{A}$ matrix is unitary and are of no direct relevance to our calculations. Additionally, *a* is an appropriate renormalization factor of the matrix $\hat{A}$. In what follows, we will use the spectral norm, i.e., the largest singular value, $||\hat{A}||=max\left\{\sigma_{A}\right\}$. Then, the application of ${\hat{U}}_{A}$ on the composite state $|0^{⊗m}\rangle |\psi \rangle$ has the orthogonal decomposition(40)$${\hat{U}}_{A}|0^{⊗m}\rangle |\psi \rangle =\frac{1}{a}|0^{⊗m}\rangle \hat{A}|\psi \rangle +|⊥\rangle ,$$
where $\langle ⊥|0^{⊗m}\rangle =0$. As with the Hadamard product implementation in Figure 1 and Figure 2, a successful projective measurement $\hat{P}=|0^{⊗m}\rangle \langle 0^{⊗m}|⊗{\hat{1}}_{2^{n}\times 2^{n}}$, in the first register with respect to the 0-bit results in the normalized state $\frac{\hat{A}|\psi \rangle }{\left||\hat{A}|\psi \rangle |\right|}$ with success probability $p_{success}=\frac{{\left||\hat{A}|\psi \rangle |\right|}^{2}}{a^{2}}$. The respective quantum circuit for this post-selective process is depicted in Figure 5.

Of the various dilation techniques [38,39,40,41,42] for ${\hat{U}}_{A}$, we employ an m=1 block encoding constructed via classical Singular Value Decomposition (SVD), combined with the LCU method [39]. This selection is motivated because the non-trivial sub-matrices ${\hat{A}}_{1}$ and ${\hat{A}}_{2}$ in Equations (Equation 9) and (Equation 14) are of small finite dimension. This leads to a classical SVD implementation complexity of $O(8^{3})$ and $O(10^{3})$, for the 1nd and 2nd order schemes, respectively. Therefore, the SVD decomposition of ${\hat{A}}_{1}$ and ${\hat{A}}_{2}$ in Equations (Equation 9) and (Equation 14) can be performed efficiently using a classical computer as part of a preprocessing routine.

The SVD decomposition of the matrix $\hat{A}$ is $\hat{A}={\hat{V}}^{†}\hat{\Sigma }\hat{W}$ where $\hat{V}$ and $\hat{W}$ are unitary matrices and $\hat{\Sigma }=diag\left(\sigma \right)$ is a diagonal matrix containing the non-negative singular values $\sigma_{i}\geq 0$. It can then be split into the sum of two unitary terms,(41)$$\hat{A}=\frac{a}{2}{\hat{V}}^{†}{\hat{\Sigma }}_{+}\hat{W}+\frac{a}{2}{\hat{V}}^{†}{\hat{\Sigma }}_{-}\hat{W},$$
where $a=||\hat{A}||$, and,(42)$${\hat{\Sigma }}_{\pm }=diag\left(e^{\pm i\theta }\right),\;\theta =\arccos\bar{\sigma },\;0<{\bar{\sigma }}_{i}\leq 1.$$

By applying the LCU technique [36] to implement the two-term unitary sum in Equation (Equation 41), only one extra qubit (m=1) is required to provide a $\left(1,||\hat{A}||\right)$-block encoding ${\hat{U}}_{A}$,(43)$${\hat{U}}_{A}=\left(\hat{H}⊗{\hat{V}}^{†}\right){\hat{U}}_{select}\left(\hat{H}⊗\hat{W}\right),$$
where ${\hat{U}}_{prep}^{LCU}=\hat{H}$ is the Hadamard gate and ${\hat{U}}_{select}^{LCU}$ is a diagonal matrix representing nested two-level *z*-rotations,(44)$${\hat{U}}_{select}=|0\rangle \langle 0|⊗{\hat{\Sigma }}_{+}+|1\rangle \langle 1|⊗{\hat{\Sigma }}_{-}={\hat{R}}_{z}.$$
As a result, the explicit quantum decomposition for the SVD-LCU block encoding ${\hat{U}}_{A}$ is presented in Figure 6.

For an arbitrary matrix $\hat{A}$, the decomposition cost of the SVD-LCU dilation within single qubit gates and CNOTs is $O[{\left(n+1\right)}^{2}2^{2n+1}]$ [39]. However, the Lorenz system is finite-dimensional, requiring 3 and 4 qubits to encode the first and second-order discretization schemes in Equations (Equation 7) and (Equation 12), respectively. Thus, the seemingly prohibitive exponential scaling becomes a constant, resulting in $O(2^{11})$ and $O(5^{2}2^{9})$ elementary gates, respectively.

### 3.3. Quantum Circuit Implementation of the Second-Order Time-Marching Scheme for the Lorenz Equations

The single-step quantum evolution $|\psi_{n}\rangle \to |\psi_{n+1}\rangle$ as outlined in the Table 1, for the second-order discretization scheme of the Lorenz model (Equations (Equation 12)–(Equation 19)) is given by(45)$${\hat{U}}_{A_{2}}{\hat{U}}_{nl}{|0\rangle }_{e}|{0^{⊗2}\rangle }_{LCU}|{\psi_{n}^{⊗2}\rangle }_{c}|\psi_{n}\rangle ={|0\rangle }_{e}|{0^{⊗2}\rangle }_{LCU}{|0000\rangle }_{c}^{⊗2}|\psi_{n+1}\rangle +{|⊥\rangle }_{e,LCU,c},$$
with 〈⊥|e,LCU,c(|0〉e|0⊗2〉LCU|0000〉c⊗2)=0.
The second-order Lorenz system, Equations (Equation 12)–(Equation 19), can be represented using 4 qubits and the amplitude encoding in Equation (Equation 20). Thus, any state $|\psi_{n}\rangle$ is a 4-qubit quantum state.The nonlinear state $|\psi_{n}^{nl}\rangle$ in Equation (Equation 13) consists of polynomial terms up to third degree in the amplitudes $x_{n}$. Subsequently, as discussed in Section 3.1.2, implementing $|\psi_{n}^{nl}\rangle$ requires a sum of Hadamard products between three copies of $|\psi_{n}\rangle$ interleaved with amplitude alternating $\hat{G}$ gates,(46)$${\hat{U}}_{nl}=\sum_{j=1}^{3}{\hat{U}}_{h}^{j}{\hat{G}}_{j}.$$The ${\hat{G}}_{j}$ operators are tensor-fold gates acting on the 12-qubit composite space of the $|\psi^{⊗3}\rangle$ states, with ${\hat{G}}_{j}={\hat{G}}_{2j}⊗{\hat{G}}_{1j}⊗{\hat{G}}_{0j}$.To implement ${\hat{U}}_{nl}$ we define a copy register (c) that stores the two additional copied states $|{\psi_{n}^{⊗2}\rangle }_{c}$ where the ${\hat{U}}_{h}^{j}$ gates act and an LCU register (LCU) to sum the respective terms, as in Figure 3 and Figure 4. The resulting evolution is(47)$${\hat{U}}_{nl}|{0^{⊗2}\rangle }_{LCU}|{\psi_{n}^{⊗2}\rangle }_{c}|\psi_{n}\rangle =|{0^{⊗2}\rangle }_{LCU}{|0000\rangle }_{c}^{⊗2}|\psi^{nl}\rangle +{|\tilde{⊥}\rangle }_{LCU,c}.$$Finally, by augmenting the nonlinear preparation process in Equation (Equation 47) with the single qubit evolution register (e), and applying the two-level variant of the block-encoding matrix ${\hat{U}}_{A_{2}}$ from Equation (Equation 43) that acts between (e) and the target register, we obtain(48)$${\hat{U}}_{A_{2}}\left({|0\rangle }_{e}{|0^{⊗2}\rangle }_{LCU}{|0000\rangle }_{c}^{⊗2}|\psi^{nl}\rangle +{|0\rangle }_{e}|\tilde{⊥}\rangle \right)={|0\rangle }_{e}|{0^{⊗2}\rangle }_{LCU}{|0000\rangle }_{c}^{⊗2}|\psi_{n+1}\rangle +{|⊥\rangle }_{e,LCU,c}.$$
Figure 7 illustrates the quantum circuit implementation corresponding to the aforementioned tasks for the second-order, single-step, time advancement of the Lorenz dynamics.

We will now extend the single-time-step evolution into a full time-marching quantum solver. In order to implement the next time step $|\psi_{n+1}\rangle \to |\psi_{n+2}\rangle$ (all states are normalized after a successful measurement) it is required to act with ${\hat{U}}_{nl}$ to prepare the intermediate nonlinear state $|\psi_{n+1}^{nl}\rangle$,(49)$${\hat{U}}_{c}{|0000\rangle }_{c}^{⊗2}|\psi_{n+1}\rangle ={|\psi_{n+1}\rangle }_{c}^{⊗2}|\psi_{n+1}\rangle \overset{{\hat{U}}_{nl}}{\to }{|0000\rangle }_{c}^{⊗2}|\psi_{n+1}^{nl}\rangle .$$
However, due to the quantum no-cloning theorem [37], no such operator ${\hat{U}}_{c}$ exists that can satisfy Equation (Equation 49) because the state $|\psi_{n+1}\rangle$ is unknown.

Subsequently, to perform the nonlinear transformation $|\psi_{n+1}\rangle \to |\psi_{n+1}^{nl}\rangle$, multiple copies of the initial state $|\psi_{n}\rangle$ must be prepared and evolved in a parallel. Therefore, based on the single-step implementation n→n+1 with ${\hat{U}}_{1}={\hat{U}}_{A_{2}}{\hat{U}}_{nl}$, the evolution operator ${\hat{U}}_{j}$ that defines the *j*-step advancement, ${\hat{U}}_{j}|\psi_{n}\rangle \to |\psi_{n+j}\rangle$ can only be defined recursively,(50)$${\hat{U}}_{j}={\hat{U}}_{1}{\hat{U}}_{j-1}^{⊗3}.$$
Indeed,(51)$${\hat{U}}_{j}{|0\rangle }_{e}|{0^{⊗2}\rangle }_{LCU}|{\psi_{n}^{⊗2}\rangle }_{c}|\psi_{n}\rangle \to {\hat{U}}_{1}{|0\rangle }_{e}|{0^{⊗2}\rangle }_{LCU}|{\psi_{n-j-1}\rangle }_{c}^{⊗2}|\psi_{n-j-1}\rangle \to |\psi_{n+j}\rangle .$$

Taking into consideration the no-cloning theorem and Equation (Equation 50), a quantum time-marching algorithm is given in Table 2.

The respective quantum circuit implementation for the time-marching algorithm in Table 2 is presented in Figure 8. A characteristic feature of the algorithm is that the copies of the initial state $|\psi_{n}^{⊗2}\rangle$ in the (c) registers are not discarded after each time step evolution. Instead, they are re-prepared using the $C{\hat{U}}_{\psi_{n}}^{⊗2}$ operation such that the (c) registers contain a sufficient number of $|\psi_{n}\rangle$ states to be used for creating the $|{\psi_{n+j-1}\rangle }_{c}^{⊗2}$ in the least significant of the (c) registers. Then, applying ${\hat{U}}_{1}$ on the $|{\psi_{n+j-1}\rangle }_{c}^{⊗2}|{\psi_{n+j-1}\rangle }_{\tau }$ state we obtain the *j*-time update. We build upon this logic by schematically presenting the re-usage of the copy register for a two-step, n→n+2, advancement following Figure 8: (52)$$\begin{array}{cc} \; & \cdots |{\psi_{n}\rangle }_{c}^{⊗2}\cdots |{\psi_{n}\rangle }_{c}^{⊗2}\underset{\mathrm{target}\;\mathrm{state}}{\underbrace{{|\psi_{n}\rangle }_{\tau }}} \end{array}$$(53)$$\begin{array}{cc} \overset{{\hat{U}}_{1}}{\to } & \cdots {|\psi_{n}\rangle }_{c}^{⊗2}\cdots \underset{\mathrm{re}\text{-}\mathrm{use}}{\underbrace{{|0000\rangle }_{c}^{⊗2}}}|{\psi_{n+1}\rangle }_{\tau }+\cdots \end{array}$$(54)$$\begin{array}{cc} \overset{C{\hat{U}}_{\psi_{n}}^{⊗2}}{\to } & \cdots |{\psi_{n}\rangle }_{c}\underset{\mathrm{available}\;\mathrm{copies}}{\underbrace{|{\psi_{n}\rangle }_{c}\cdots |{\psi_{n}\rangle }_{c}}}\;\underset{\mathrm{intermediate}\;\mathrm{target}\;\mathrm{state}}{\underbrace{{|\psi_{n}\rangle }_{c}}}{|\psi_{n+1}\rangle }_{\tau }+\cdots \end{array}$$(55)$$\begin{array}{cc} \overset{{\hat{U}}_{1}}{\to } & \cdots |{\psi_{n}\rangle }_{c}\underset{\mathrm{re}\text{-}\mathrm{use}}{\underbrace{|0000\rangle \cdots |0000\rangle }}|{\psi_{n+1}\rangle }_{c}|{\psi_{n+1}\rangle }_{\tau }+\cdots \end{array}$$(56)$$\begin{array}{cc} \overset{C{\hat{U}}_{\psi_{n}}^{⊗2}}{\to } & \cdots \underset{\mathrm{available}\;\mathrm{copies}}{\underbrace{{|\psi_{n}\rangle }_{c}^{⊗2}}}\cdots \underset{\mathrm{intermediate}\;\mathrm{target}\;\mathrm{state}}{\underbrace{{|\psi_{n}\rangle }_{c}}}{|\psi_{n+1}\rangle }_{c}{|\psi_{n+1}\rangle }_{\tau }+\cdots \end{array}$$(57)$$\begin{array}{cc} \overset{{\hat{U}}_{1}}{\to } & \cdots {|0000\rangle }_{c}^{⊗2}\cdots \underset{\mathrm{evolution}\;\mathrm{to}\;n+2}{\underbrace{|{\psi_{n+1}\rangle }_{c}^{⊗2}|{\psi_{n+1}\rangle }_{\tau }}}+\cdots \end{array}$$(58)$$\begin{array}{cc} \overset{{\hat{U}}_{1}}{\to } & \cdots {|0000\rangle }_{c}^{⊗2}\cdots {|0000\rangle }_{c}^{⊗2}|{\psi_{n+2}\rangle }_{\tau }+\cdots \end{array}$$
This re-usage of qubits [19] allows for parallel processing of the quantum states and optimal scaling in the numbers of required copies of the state $|\psi_{n}\rangle$, as will be discussed in the next section.

### 3.4. Complexity Considerations and Discussion

In quantum algorithmic architectures incorporating many copies of the initial state $|\psi_{n}\rangle$, the important aspects dictating the efficiency of the algorithm are the number of copies that the algorithm requires (circuit width) and the number of elementary operations (circuit depth).

Due to the re-usage of the states in the (c) registers, as presented in Equations (Equation 53)–(Equation 58), the required number of copies, $N_{copies}$, of the 4-qubit state $|\psi_{n}\rangle$ scales linearly with respect to the number of time integration steps $N_{t}$ for a simulation $n\to n+N_{t}$,(59)$$N_{copies}\left(N_{t}\right)=4N_{t}-1.$$
This linear dependence of the number of copies on $N_{t}$ establishes an exponential improvement over the results of Ref. [15] and a quadratic improvement compared to Refs. [11,16]. This technique was first demonstrated in Ref. [19]. Taking into consideration the ancillary qubits in the (t)(LCU) and (e) registers, the total number of working quantum resources is(60)$$N_{qubits}\left(N_{t}\right)=\left(2N_{t}-1\right)\left(\underset{\left(c\right)}{\underbrace{8}}+\underset{\left(e\right)}{\underbrace{1}}+\underset{\left(LCU\right)}{\underbrace{2}}\right)+\underset{\left(t\right)}{\underbrace{\log_{2}N_{t}+1}}+\underset{\mathrm{target}}{\underbrace{4}}=22N_{t}+\log_{2}N_{t}-6.$$
Therefore, the total number of operational qubits for the overall quantum time-integration scheme scales as $N_{qubits}\left(N_{t}\right)=O\left(N_{t}\right)$.

On the other hand, the recursive structure of the time-marching algorithm in Table 2 enables a direct count of the operators ${\hat{U}}_{1}$, $C{\hat{U}}_{\psi_{n}}^{⊗2}$ and $\hat{S}$ which are highlighted in blue, yellow and red respectively in Figure 8. Specifically, the number $N_{U_{1}}$ of the ${\hat{U}}_{1}$ gates that comprise each ${\hat{U}}_{j}$ is(61)$$N_{U_{1}}\left(j\right)=3^{j-1}.$$
Thus, the total number of ${\hat{U}}_{1}$ gates for $N_{t}$ time-steps reads(62)$$N_{U_{1}}\left(N_{t}\right)=\sum_{j=1}^{N_{t}}3^{j-1}=\frac{3^{N_{t}}}{2}.$$
Accordingly, the number of $\hat{S}$ gates (in red in Figure 8) acting on the clock (t) register is $N_{S}\left(N_{t}\right)=2N_{t}$ and, similarly with Equation (Equation 62) due to recurrence, the number of the ${\hat{U}}_{\psi_{n}}^{⊗2}$ gates (in yellow in Figure 8) scales as(63)$$N_{U_{\psi_{n}}^{⊗2}}\left(N_{t}\right)=3^{N_{t}-1}-N_{t}.$$
As a result, the total number of queries to the building blocks of our algorithm is(64)$$N_{queries}=\frac{5}{6}3^{N_{t}}+N_{t}=O\left(3^{N_{t}}\right).$$
Finally, since a second-order time-discretization scheme has been employed, the total simulation error ε in terms of the total simulation time *T* is $\varepsilon =T^{2}/N_{t}$.

The decomposition cost of the ${\hat{U}}_{1}$ and $C{\hat{U}}_{\psi_{n}}$ operators into elementary gates is constant because each of the ${\hat{U}}_{nl}$, ${\hat{U}}_{A_{2}}$ and ${\hat{U}}_{\psi_{n}}$ operators act in the 4-qubit space (see Section 3.1.1, Section 3.1.2 and Section 3.2). The corresponding circuit implementation has been explicitly provided in Figure 3, Figure 6 and Figure 7. Thus, the resulting implementation scaling into CNOTs and single-qubit gates scales as $O\left(28\cdot 2^{9}N_{t}^{2}\right)=O\left(60\cdot 16^{2}\right)$. Implementation of the $\hat{S}$ operator scales as $O\left(\nu^{2}\right)=O\left(\log_{2}^{2}N_{t}\right)$.

Consequently, the proposed quantum solver for the second-order discretized Lorenz system requires an operational circuit width that scales linearly with the time-integration steps and with overall complexity,(65)$$O\left(60d^{2}p^{N_{t}},T^{2}/\epsilon \right),\;p=3,\;d=16,\;N_{copies}\left(N_{t}\right)=4N_{t}-1.$$

Notice that d=16 is the dimension of the second-order discretized Lorenz system in Equation (Equation 13), and p=3 is the maximum polynomial degree in the nonlinear state $|\psi_{n}^{nl}\rangle$.

The complexity scaling presented in Equation (Equation 65) is characteristic of the proposed quantum algorithmic scheme. Consider a polynomial nonlinear system of ODEs of dimension $d_{s}$ with a maximum degree $p_{s}$. Then, employing a *K*-th order discretization scheme, the resulting nonlinear state $|\psi^{nl}\rangle$ has a polynomial degree of $p=p_{s}+K-1$ and dimension *d*. Therefore, the augmented system (similar to that of Equations (Equation 7) and (Equation 12)) can be represented within $n=\log_{2}d$ qubits. To prepare the state $|\psi^{nl}\rangle$ through the Hadamard product we need *p* copies. Therefore, the complexity of the general high order quantum solver in analogy with Equation (Equation 65) is given by(66)$$O\left[4^{n}n^{2}p^{N_{t}},{\left(\frac{T^{K}}{\varepsilon }\right)}^{\frac{1}{K-1}}\right]=O\left[d^{2}\log_{2}^{2}dp^{N_{t}},{\left(\frac{T^{K}}{\varepsilon }\right)}^{\frac{1}{K-1}}\right],\;N_{copies}\left(N_{t}\right)=O\left(N_{t}\right).$$
When the resulting ${\hat{A}}_{K}$ matrix is *d*-sparse, the scaling with respect the dimension of the original nonlinear system is $O[poly\log\left(d_{s}\right)p^{N_{t}}]$. As a result, the quantum solver exhibits an exponential speed-up over classical ODE solvers, which typically scale linearly with the dimension $d_{s}$. However, as with all contemporary quantum algorithms for polynomial nonlinear ODEs, the present quantum solver exhibits an exponential lower bound in its time complexity (see, for example, Ref. [43], where the authors develop a quantum algorithm for nonlinear and norm-preserving ODEs). In contrast, classical Runge–Kutta solvers achieve polynomial complexity, $O[poly\left(N_{t}\right)]$, with respect to the number of integration steps [44].

The novelty of the complexity scaling in Equation (Equation 66) compared to other time-marching schemes, lies in its linear dependence on the number of state copies with respect to the integration steps $N_{t}$, even when employing higher-order explicit integration schemes rather than the simple Euler method used in previous studies. Using quantum compression gadgets [45,46], it is anticipated that the above scalings in Equations (Equation 59)–(Equation 66) can be further improved.

The final feature of the algorithm to be discussed is its post-selective, and therefore, probabilistic nature. As illustrated in Figure 8, obtaining the correct result $|\psi_{n+N_{t}}\rangle$ requires a single successful measurement at the output in all ancillary registers with respect to the 0-bits. While the continuum Lorenz nonlinear differential equations yield a contraction map in time, as shown in Equation (Equation 5), this does not hold for the discretized Lorenz system, i.e., for the operator ${\hat{A}}_{2}$ in Equation (Equation 14). As a result, a normalization factor $a=\sigma_{max}$, where $\sigma_{max}$ is the maximum singular value of ${\hat{A}}_{2}$, dictates the success probability of the SVD-LCU block encoding ${\hat{U}}_{A_{2}}$, as described in Section 3.2. Since ${\hat{U}}_{A_{2}}$ is a component in the fundamental algorithmic composite block ${\hat{U}}_{1}={\hat{U}}_{A_{2}}{\hat{U}}_{nl}$ in Equation (Equation 45), and the number of ${\hat{U}}_{1}$ repetitions scale according to Equation (Equation 62), then the overall success implementation probability is(67)$$p_{success}\left(N_{t}\right)\propto {\left(\frac{1}{\sigma_{max}^{2}}\right)}^{3^{N_{t}}}\approx {\left(1-2\varepsilon_{\sigma }\right)}^{3^{N_{t}}},\;\;\mathrm{with}\;\;\sigma_{max}=1+\varepsilon_{\sigma },\;\varepsilon_{\sigma }>0.$$

In turn, the value of $\sigma_{max}$, for the ${\hat{A}}_{2}$ matrix depends on the chosen time-discretization step δt as dictated in Equations (Equation 15)–(Equation 19). The behavior of $\sigma_{max}\left(\delta t\right)$ as a function of the time step δt is shown in Figure 9. Selecting a time step δt<<1 improves the accuracy of the algorithm and results in a $\sigma_{max}<1.01$ according to Figure 9. Thus, the implementation success probability in Equation (Equation 67) is improved. However, choosing an extremely small δt<<1, it increases the number of iteration steps $N_{t}$ for a total simulation time $t\to t+T,\;\;T=N_{t}\delta t$. Because of the doubly exponential dependence of the success probability on $N_{t}$ in Equation (Equation 67), this leads to a significant reduction in the overall probability for a successful implementation of our quantum algorithm.

The issue of vanishing implementation probability also arises in time-marching algorithms for dissipative and linear ODEs [46]. Consequently, at the output of the algorithm, probability amplification techniques such as amplitude amplification [47], oblivious amplitude amplification [48] and uniform singular value amplification [46] are required to increase the success probability of measuring the state with all ancillary qubits in the 0-bit.

We close this section by highlighting that even after successfully post-selecting the state $|\psi_{n+N_{t}}\rangle$, the information about the solution vector $\left(x_{n+N_{t}},y_{n+N_{t}},z_{n+N_{t}}\right)$ remains encoded in the amplitudes of $|\psi_{n+N_{t}}\rangle$, according to Equation (Equation 20). Subsequently, to retrieve the solution, it is necessary to apply amplitude estimation [47], which worsens the exponential time scaling of the algorithm. An alternative approach is to construct a time history state by creating the entangled state,(68)$$|\psi_{h}\rangle =\frac{1}{\sqrt{N_{t}+1}}\sum_{j=0}^{N_{t}}|j\rangle |\psi_{n}\rangle ,$$
and, upon applying the operator,(69)$${\hat{U}}_{history}=\sum_{j=0}^{N_{t}}|j\rangle \langle j|⊗{\hat{U}}_{j},\;{\hat{U}}_{j}|\psi_{n}\rangle =|\psi_{n+j}\rangle ,$$
yields a time history state comprising all intermediate solutions within $\left[0,N_{t}\right]$,(70)$${\hat{U}}_{history}|\psi_{h}\rangle =\frac{1}{\sqrt{N_{t}+1}}\sum_{j=0}^{N_{t}}|j\rangle |\psi_{n+j}\rangle .$$
The derivation in Equation (Equation 70) has been simplified in the sense that each |*j*〉 is entangled with unwanted contributions from the action of ${\hat{U}}_{j}$. However, by defining a physically meaningful observable (Hermitian operator) $\hat{D}$, we can compute its mean value $J_{D}$,(71)$$J_{D}=\langle \hat{D}\rangle =\frac{1}{N_{t}+1}\sum_{j=0}^{N_{t}}\langle \psi_{n+j}|\hat{D}|\psi_{n+j}\rangle \overset{\mathrm{continuous}\;\mathrm{limit}}{\to }\frac{1}{T}\int_{0}^{T}\langle \psi (t)|\hat{D}|\psi (t)\rangle d\;t,$$
where $J_{D}$ is the instantaneous average of a physically interesting quantity such as the finite-time Lyapunov exponent [49] or other fluid dynamics related quantities [50]. In general, for chaotic and turbulent systems, especially in the infinite average time limit $N_{t}\to \infty$, the classical cost of an accurate computational estimate of $J_{D}$ is prohibitive [50]. Lastly, the time history state (Equation 70) can serve as the input for quantum data processing, enabling dynamical analysis in terms of principal components, power spectra, and wavelet decompositions [51].

## 4. Numerical Demonstrations

While the proposed quantum solver exhibits a quantum speed-up in terms of the dimension of the system compared to the classical ODE solvers, the exponential scaling in the number of operations with the integration steps in Equation (Equation 65) puts limitations to the quantum implementation of our algorithm in the Noisy Intermediate-Scale Quantum (NISQ) era, with relatively high error rates and relatively small number of allowed operations. Therefore, we resort to classical implementation of the quantum algorithm presented in Section 3.2 for the Lorenz second-order scheme, Equations (Equation 12)–(Equation 19), for various values of β at σ=10 and ρ=28.

Following the work of Moysis [52], we plot the bifurcation diagram, Figure 10, for z(β) as a function of the parameter β in the range 0.54<β<0.58 with σ=10 and ρ=28. This bifurcation diagram is a Poincare plot of the Lorenz attractor as its orbit intersects the plane x=0 with dx/dt<0. One can readily identify the bifurcation from period-1 (P1) to period-2 (P2) limit cycle for β≈0.542. A bifurcation from P2 - $P2^{2}$ around β≈0.558, and a $P2^{2}\;\text{-}\;P2^{3}$ bifurcation for β≈0.5624. Indeed, in this small parameter window 0.54<β<0.58, this bifurcation diagram is formally similar to the well-known period-doubling bifurcation route to chaos for the 1D discrete logistic map $\eta_{n+1}=\mu \eta_{n}\left(1-\eta_{n}\right)$ as the parameter μ is varied.

In the following, we will compare the Lorenz regular (limit cycles) and chaotic attractors as determined from our second-order fixed time scale solution to Equations (Equation 12)–(Equation 19) with those derived from the higher order adaptive time scale ODE solver in Mathematica. For values of β giving rise to a chaotic attractor (e.g., β=0.58), the second-order finite difference scheme will require a smaller time step $\delta t=2.5\times 10^{-4}$ than is needed to determine the periodic limit cycles (where $\delta t=10^{-3}$).

### 4.1. Chaotic Attractor: β=0.58

The β-bifurcation diagram, Figure 10, indicates that we should expect chaos for β=0.58. Indeed, we find such an attractor using the higher order adaptive time ordinary differential equation (ODE) solver, as presented in Figure 11. The asymmetry in the lobes makes this attractor look considerably different than the standard picture of the “butterfly” attractor for the more common parameter value β=8/3. Here, both the 2D phase plot in the y−z plane as well as the full 3D attractor in the x−y−z plane for initial conditions x(0)=0.1, y(0)=−1.1, z(0)=10.1 are presented in Figure 11. Notice that, by changing just the initial *z* to z(0)=1.1, but keeping x(0)=0.1 and y(0)=−1.1, the resulting Lorenz attractor is still chaotic, but with mirror symmetry.

Implementing the quantum solver for the second-order finite difference scheme, we find excellent qualitative agreement between the second-order-generated chaotic attractor in Figure 12 and that generated by the adaptive grid higher order ODE solver used in Figure 11. This signifies that a successful quantum implementation of our method has the potential to resolve the chaotic region for the Lorenz system, preserving the basic structural characteristics of the chaotic attractor.

### 4.2. Limit Cycles

Aside from the chaotic attractors, different structures that arise in the region of regular dynamics of the Lorenz system are limit cycles. Limit cycles are isolated periodic orbits, which also exhibit sensitivity to perturbations. For example, in Figure 10, as the parameter β increases, there is a transition through bifurcations from the regular dynamics limit cycles to chaos.

In the following, some of the period-doubling non-chaotic Lorenz attractors are considered, with initial conditions x(0)=0.1, y(0)=−1.1, z(0)=1.1, using the second-order scheme.

#### 4.2.1. P1-Limit Cycle: β=0.52

The period-1 limit cycle for β=0.52 is shown in Figure 13, anchored about the two unstable fixed points (±3.75,±3.75,27).

#### 4.2.2. Period Doubling Limit Cycles

On increasing β, to β=0.55 we encounter a period-2 (P2) limit cycle, depicted in Figure 14, and a period-4 $\left(P2^{2}\right)$ limit cycle at β=0.56 presented in Figure 15.

The period-doubling limit cycles found in Figure 13, Figure 14 and Figure 15 with the second-order method can be considered rudimentary in terms of their period-doubling behavior as the parameter β increases toward the formation of the chaotic attractor. To further challenge the capabilities of the proposed second-order quantum integration method, we seek to verify whether we can generate a non-trivial limit cycle. Such non-trivial limit cycles arise as transitions from chaos to regular dynamics, as indicated in the bifurcation diagram.

For instance, at β=0.5648, the Lorenz attractor is a P6 limit cycle resulting from the period-doubling bifurcation of the two P3 attractors. Such isolated and fine-structured solutions cannot be obtained by series truncation techniques such as Carleman and KvN or by variational algorithms. In contrast, our method accurately captures the structure of the P6-limit cycle in Figure 16 when compared to the Mathematica ODE solver in Figure 17.

The numerical demonstrations in Figure 11, Figure 12, Figure 13, Figure 14, Figure 15, Figure 16 and Figure 17 illustrate that the second-order discretization scheme captures the correct qualitative nonlinear Lorenz dynamics. In contrast, prior works [14,15,16,18] rely exclusively on the simple first-order Euler scheme, which is insufficient for accurately simulating the Lorenz dynamics [27]. Our algorithm offers a systematic framework for quantum implementation of second and higher-order explicit discretization schemes for reliable quantum simulation of nonlinear ODEs. Importantly, as shown in Equation (Equation 66), the quantum realization of these higher-order schemes consistently requires only a linear number of state copies. However, as discussed in Section 3.4, the quantum algorithm is not competitive in time complexity with the classical integration methods. Therefore, there is no practical motivation to quantitatively compare the second-order results to those of the higher-order adaptive Mathematica solver in terms of the $L^{2}$ error.

## 5. Conclusions

In the present paper, we present a quantum algorithm that belongs to the class of quantum time-marching solvers and apply it to a second-order discretized Lorenz system of nonlinear differential equations. For a single-time-step update, a two-stage process is required: first, the preparation of a nonlinear quantum state, and then the linear evolution of the state. To construct the nonlinear 4-qubit Lorenz state $|\psi_{n}^{nl}\rangle$, the Hadamard product is employed, whereas the action of the linear non-unitary operator ${\hat{A}}_{2}$ is performed through a simple SVD-LCU block encoding technique. Extending the single-time-step evolution into a complete time-marching scheme requires evolving multiple copies of the initial state $|\psi_{n}\rangle$ in parallel due to the no-cloning theorem. By incorporating a re-usage of states [19] strategy, our algorithm obtains a recursive structure that allows for linear scaling in the number of state copies with respect to the number of iteration steps $N_{t}$ (Equation (Equation 59)). This represents an improvement over other quantum time-marching algorithms, which require exponential or quadratic resource scaling.

Additionally, the novelty of the algorithm can be summarized as follows: (1) it is explicit with respect to the gate decompositions (no oracle operations) and avoids any linearization techniques such as Carleman or KvN by providing a path for accurate simulation of system dynamics; (2) it is pertinent for developing higher order discretization schemes relevant to numerical implementation of stiff ODEs, such as the Lorenz system; (3) in spite of the exponential time scaling, the proposed quantum solver exhibits a quantum speed-up with respect to system dimensionality $d_{s}$, as implied by Equation (Equation 65)—this is particularly valuable for densely discretized partial differential equations (PDEs), resulting in huge-dimensional ODEs for resolving the multiscale structures in turbulent flows. The recursive structure of the proposed algorithmic scheme has been identified as a promising feature for developing novel quantum algorithms with a quantum advantage [53].

Classical implementation of the proposed quantum algorithm for the second-order discretized Lorenz system demonstrates strong agreement with the Mathematica-generated chaotic and regular attractors. Combined with the improved quantum resources scaling presented in Equation (Equation 60) and the aforementioned points (1)–(3), these results suggest that the proposed time-marching quantum algorithm for discretized nonlinear differential equations has the potential to enable significant advances in the quantum simulation of classical nonlinear dynamics.

It is of great interest to extend the presented ideas to PDEs by employing a second-order time-discretization scheme but now within the framework of Qubit Lattice Algorithms (QLA). A Qubit lattice algorithm consists of a sequence of interleaved unitary collide-stream operators recovering the first-order time-discretized PDE up to second-order accuracy in spatial grid size δx. In the continuum limit, it recovers the correct partial differential equation, provided the time displacement δt obeys a diffusion ordering: $\delta t\approx \delta x^{2}$. This thus leads to an essentially second-order QLA scheme.

Qubit lattice algorithms have been applied to the nonlinear Gross–Pitaevskii system of equations, which describe the ground-state dynamics of spinor Bose–Einstein condensates [54] and references therein. In particular, in one dimension, QLA was used to study the long-time evolution of both scalar and vector soliton collisions. The numerical results were compared to exact analytical solutions of the nonlinear equations, demonstrating excellent agreement, even though the time advancement scheme appears to be a simple Euler scheme.

Finally, QLA has also been employed for solving linear partial differential equations, such as Maxwell’s equations, exhibiting a quantum speed-up compared to classical finite time difference schemes [55].

## Figures and Tables

**Figure 1 entropy-27-00871-f001:**
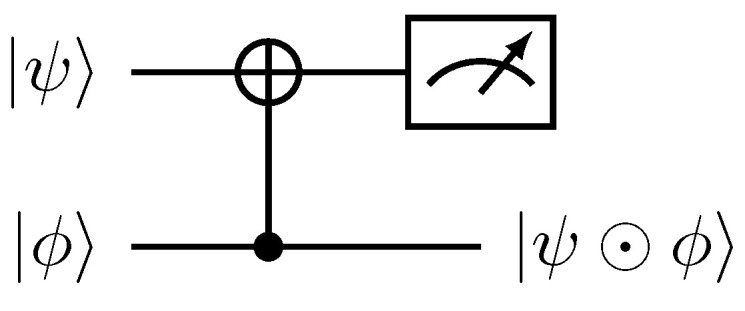
Quantum circuit implementation of the Hadamard product |ψ〉⊙|ϕ〉 between single-qubit states. A projection measurement operator $\hat{P}=|0\rangle \langle 0|⊗{\hat{1}}_{2\times 2}$ is applied to the first register for a successful implementation.

**Figure 2 entropy-27-00871-f002:**
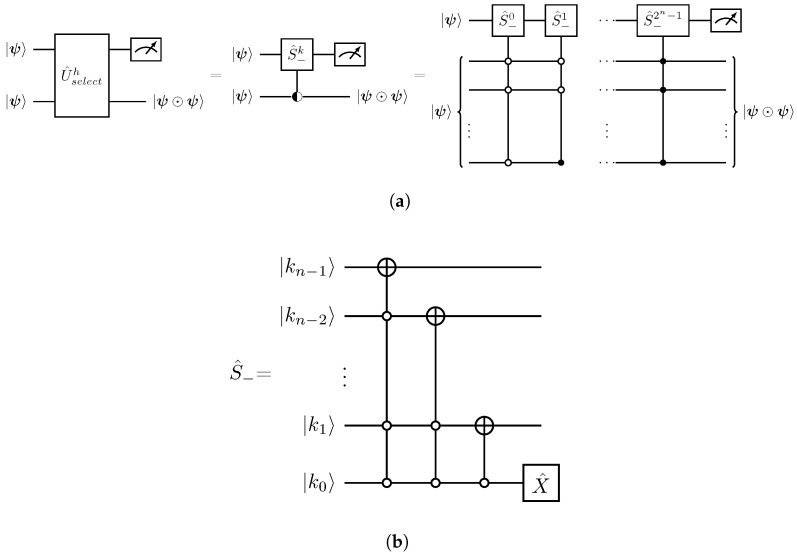
Quantum circuits for the Hadamard product between two *n*-qubit states. (**a**) Schematic implementation of the ${\hat{U}}_{select}^{h}$ operator. The symbol 
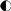
 denotes a uniform quantum multiplex gate [30,31]. (**b**) Quantum circuit implementation of the ${\hat{S}}_{-}$ operation acting on the |*k*〉 basis expressed in its binary form, $|k\rangle ={|k_{n-1}k_{n-2}\ldots k_{1}k_{0}\rangle }_{b}$. The $\hat{X}$ gate is the Pauli-x matrix.

**Figure 3 entropy-27-00871-f003:**
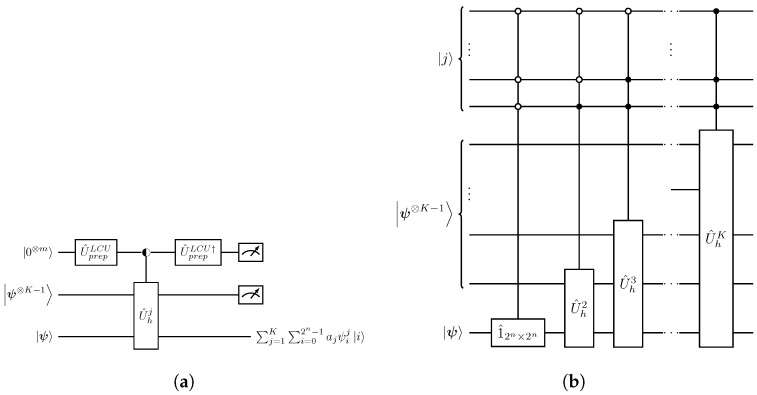
Quantum circuit implementation of the ${\hat{U}}_{nl}$ operator of Equation (Equation 30) mediating the nonlinear polynomial transformation in Equation (Equation 21). (**a**) Schematic implementation of ${\hat{U}}_{nl}$. (**b**) Explicit implementation of the uniform multiplexed operation ${\hat{U}}_{select}^{LCU}$ in Equation (Equation 32). The decomposition for each of the nj-qubit ${\hat{U}}_{h}^{j}$ operations can be derived from the respective one in Figure 2.

**Figure 4 entropy-27-00871-f004:**
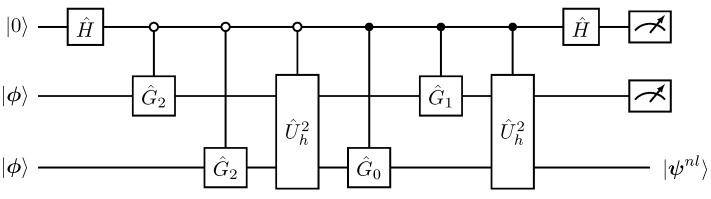
Explicit quantum circuit preparation of the nonlinear state $|\psi^{nl}\rangle$ in Equation (Equation 33). Implementation of ${\hat{U}}_{h}^{2}$ operator has been presented in Figure 2.

**Figure 5 entropy-27-00871-f005:**
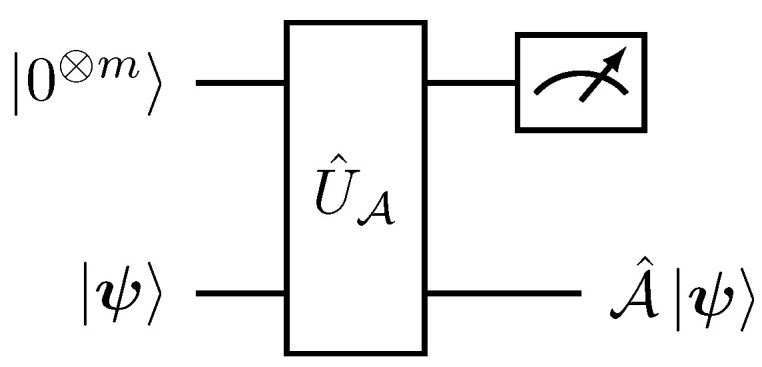
Quantum circuit implementation of the non-unitary operator $\hat{A}$ through the general block encoding ${\hat{U}}_{A}$ in Equations (Equation 39) and (Equation 40).

**Figure 6 entropy-27-00871-f006:**
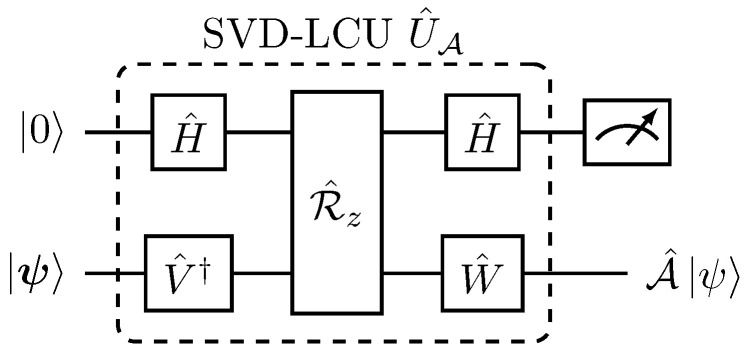
Explicit quantum circuit implementation for the SVD-LCU block encoding ${\hat{U}}_{A}$ of Equation (Equation 43).

**Figure 7 entropy-27-00871-f007:**
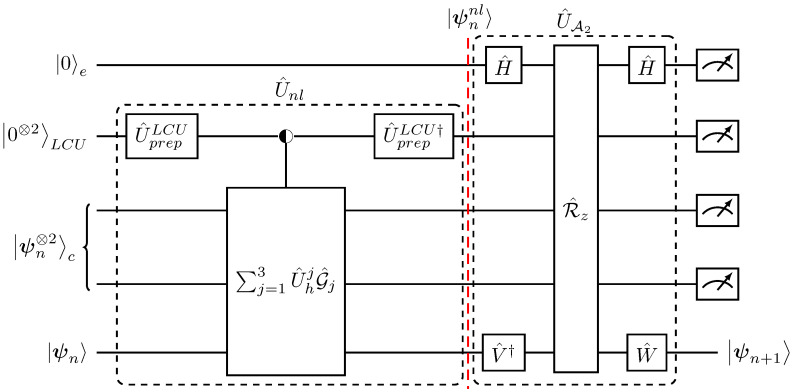
Quantum circuit implementation of the unitary two-stage process in Equation (Equation 45), corresponding to a single-time-step evolution for the second-order discretized Lorenz model. The target register is at the bottom of the circuit, where the $|\psi_{n}\rangle \to |\psi_{n+1}\rangle$ advancement is achieved.

**Figure 8 entropy-27-00871-f008:**
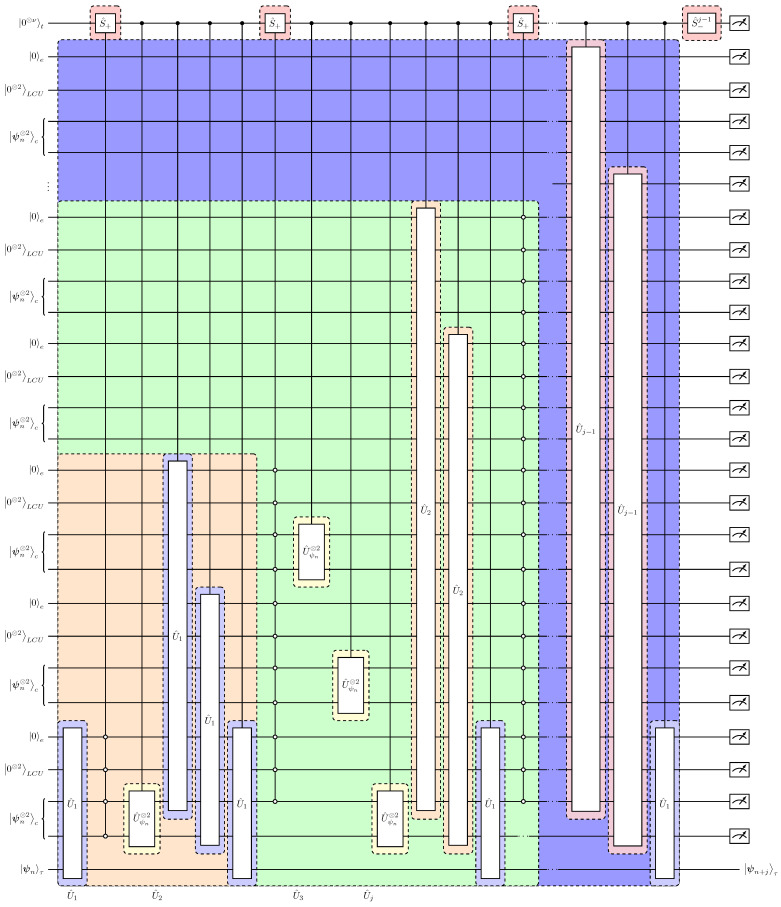
Schematic quantum circuit implementation of the time-marching quantum algorithm for the steps 1–7 in Table 2. The explicit implementation of the single-time-step evolution operator ${\hat{U}}_{1}={\hat{U}}_{A_{2}}{\hat{U}}_{nl}$ has been presented in Figure 7. The coloring reveals the recursive nested structure. For example the operator ${\hat{U}}_{3}$ (green color shade) contains 2 applications of the ${\hat{U}}_{2}$ gates and one of ${\hat{U}}_{1}$. In turn the ${\hat{U}}_{2}$ operators (orange color shade) contain 4 ${\hat{U}}_{1}$ gates. The shift $\hat{S}$ gates acting on the clock (t) register are colored in red and the preparation operators ${\hat{U}}_{\psi_{n}}^{⊗2}$ in yellow.

**Figure 9 entropy-27-00871-f009:**
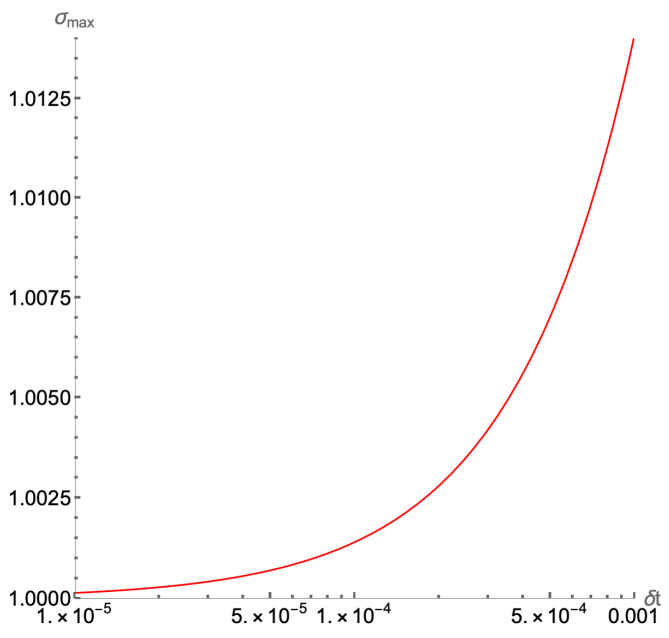
The dependence of the largest singular value of ${\hat{A}}_{2}$ matrix in Equation (Equation 14), $\sigma_{max}$, on the discretization time step δt for the typical time steps used in Section 4.

**Figure 10 entropy-27-00871-f010:**
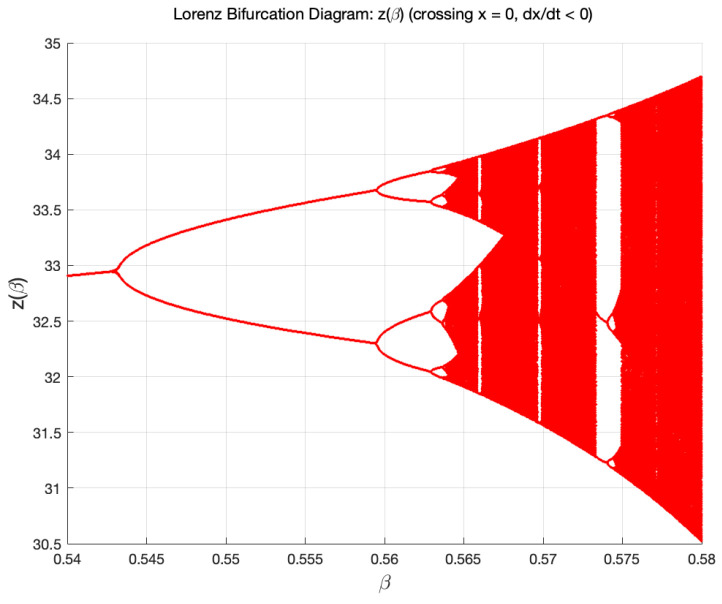
The bifurcation diagram for z(β) for 0.54<β<0.58, with σ=10, ρ=28. The bifurcation diagrams for x(β) and y(β) are qualitatively similar and have exactly the same bifurcation points in β.

**Figure 11 entropy-27-00871-f011:**
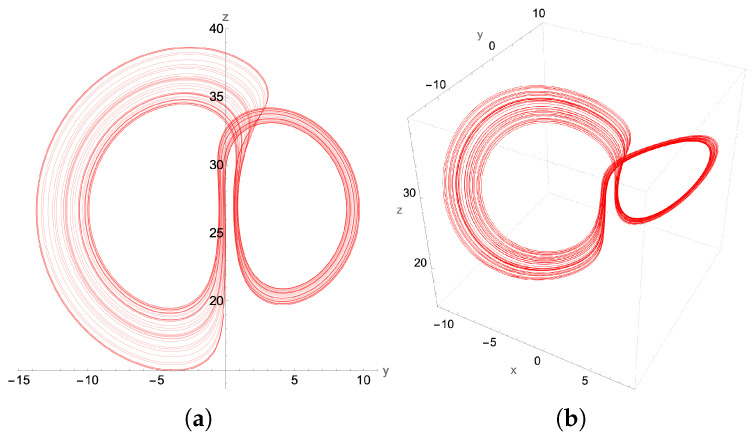
The chaotic attractor for β=0.58, σ=10 and ρ=28, showing both (**a**) its y-z projection and (**b**) the 3D plot. Initial conditions: x(0)=0.1, y(0)=−1.1, z(0)=10.1. These plots were generated by a higher order adaptive time Mathematica ODE solver. The lobes of the attractor are somewhat anchored around the two unstable fixed points of the Lorenz equations $\left(\pm \sqrt{\beta (\rho -1)},\pm \sqrt{\beta (\rho -1)},\rho -1\right)$.

**Figure 12 entropy-27-00871-f012:**
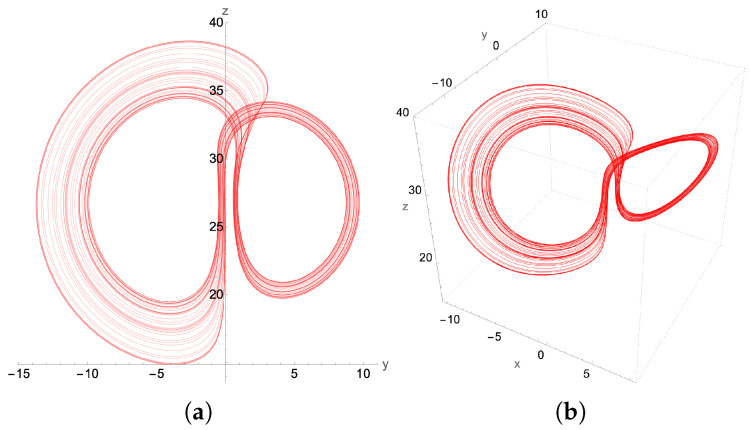
The chaotic attractor for β=0.58, and with the same initial conditions as in Figure 11. (**a**) its y-z projection and (**b**) the 3D plot. However, this attractor is determined from the classical implementation of the 2nd order recursive quantum algorithm with fixed time step $\delta t=2.5\times 10^{-4}$.

**Figure 13 entropy-27-00871-f013:**
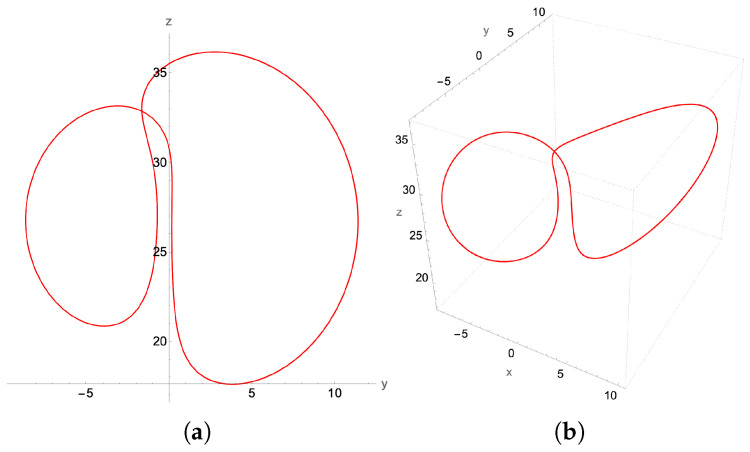
P1-limit cycle with β=0.52 and time step $\delta t=10^{-3}$: (**a**) y-z projection, (**b**) the 3D attractor.

**Figure 14 entropy-27-00871-f014:**
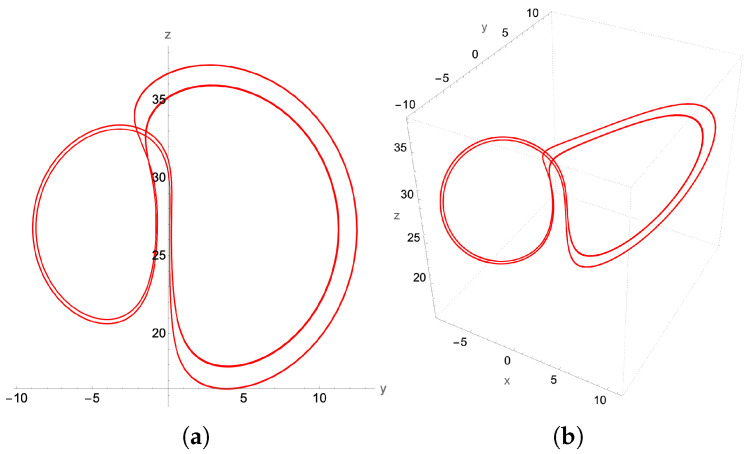
P2-limit cycle, with β=0.55 and time step $\delta t=10^{-3}$: (**a**) y-z projection, (**b**) the 3D attractor.

**Figure 15 entropy-27-00871-f015:**
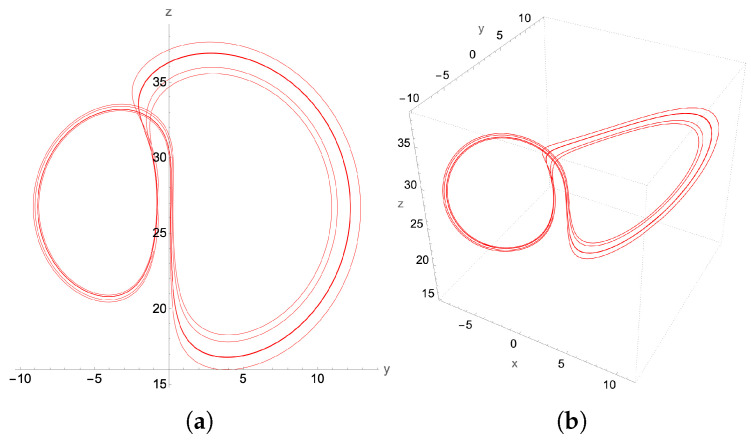
$P2^{2}$-limit cycle, with β=0.56 and time step $\delta t=10^{-3}$: (**a**) y-z projection, (**b**) the 3D attractor.

**Figure 16 entropy-27-00871-f016:**
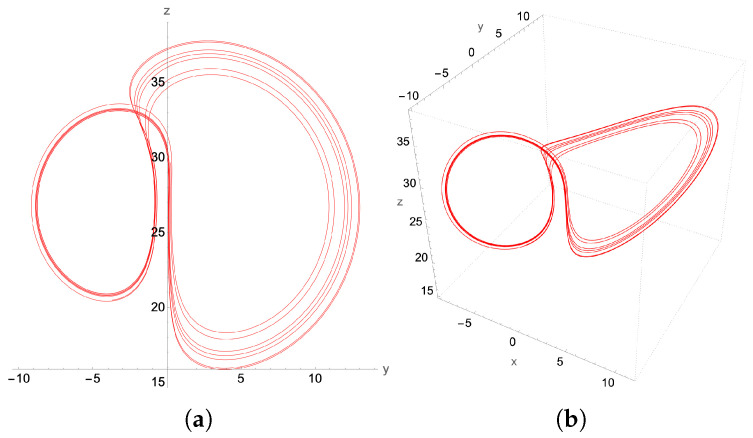
P6-limit cycle for β=0.5648 and a reduced time step $\delta t=5\times 10^{-4}$: (**a**) y-z projection, (**b**) the 3D attractor.

**Figure 17 entropy-27-00871-f017:**
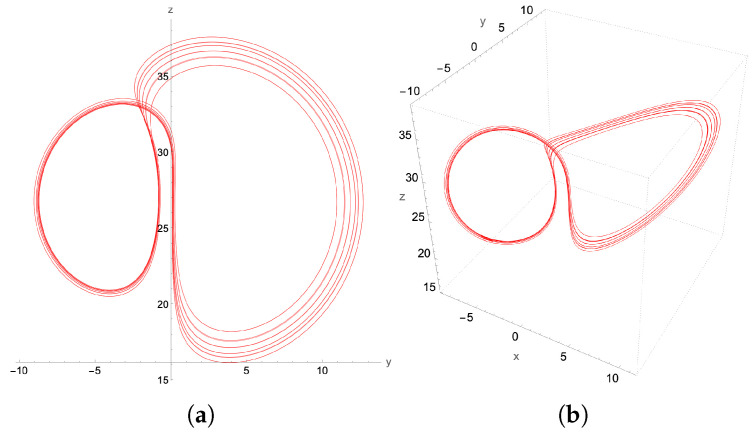
P6-limit cycle generated by a higher order adaptive time Mathematica ODE solver. (**a**) y-z projection, (**b**) the 3D attractor. Compared to the second-order scheme result in Figure 16, the main characteristics of the P6 structure are accurately captured.

**Table 1 entropy-27-00871-t001:** Operations required for a single-time-step update of Equations (Equation 7) or (Equation 12).

Single-Step Evolution n→n+1.
1. Initial state ← $\psi_{n}$
2. Preparation of the nonlinear state $\psi_{n}^{nl}$, $f\left(\psi_{n}\right)=\psi_{n}^{nl}$
3. Evolution with the time-advancement matrix $\hat{A}$, $\psi_{n+1}=\hat{A}\psi_{n}^{nl}$

**Table 2 entropy-27-00871-t002:** Quantum algorithm for the simulation of the second-order discretized Lorenz system (Equation 12).

Time-Marching Evolution n→n+j.
1. Initial target state ← ${|\psi_{n}}_{\tau }\rangle$.
2. Prepare a clock-time (t) register ← ${|0^{\nu }\rangle }_{t}$ with $\nu =[\log_{2}j]+1$ and 2j−1 copies of the (e), (LCU) and (c) registers ← ${\left({|0\rangle }_{e}|{0^{⊗2}\rangle }_{LCU}|{\psi_{n}^{⊗2}\rangle }_{c}|\psi_{n}\rangle \right)}^{⊗2j-1}$. The target state is chosen to be the least significant.
3. Evolve the target state by acting with the single-step evolution operator ${\hat{U}}_{1}$ using the least significant of the (e),(LCU),(c)-registers as dictated in Equation (Equation 45), $|\psi_{n}\rangle \to |\psi_{n+1}\rangle$.
4. Update the clock (t) register, $C{\hat{S}}^{+}{|0\rangle }_{t}\to {|1\rangle }_{t}$ to keep track of the time advancement of the target state controlled by the 0-bits in the previously used least significant, i=1 of the (e),(LCU),(c)-registers.
5. Replenish the states ${|\psi_{n}\rangle }_{c}^{⊗2}$ in the (c) register with a control operator $C{\hat{U}}_{\psi_{n}}^{⊗2}$ with respect to the 1-bit state in the (t) register with ${\hat{U}}_{\psi_{n}}|0000\rangle =|\psi_{n}\rangle$.
6. Recursively repeat j−1 times the steps 3, 4 and 5, using ${\hat{U}}_{j}$ in Equation (Equation 50) and 2j−1 ancillary registers for each iteration.
7. Initialize the clock (t) register to the ${|0\rangle }_{t}$ state and measure all registers in respect to the 0-bit states to obtain $|\psi_{n}\rangle \to |\psi_{n+j}\rangle$.

## Data Availability

The data generated in this study are available upon reasonable request from the authors.

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
