# Peer review of "Time-Marching Quantum Algorithm for Simulation of Nonlinear Lorenz Dynamics"

_entropy, 2025, doi:10.3390/e27080871_

Round 1

Reviewer 1 Report

Comments and Suggestions for Authors

A quantum algorithm is constructed for Lorenz model. To my opinion, this is a very exiting idea to combine quantum algorithms and partial differential equations. The model is chosen successfully.  On one side, it is sufficiently simple to be solved by methods of mathematical analysis and on the other side it admits an efficient implementation of quantum algorithms. Quantum algorithm is applied to a second-order discretized Lorenz system of nonlinear differential equations. It allows to use numerical methods to verify the results. I recommend to publish this paper in its present form.

Reviewer 2 Report

Comments and Suggestions for Authors

In the manuscript entropy-3752824, the authors developed a quantum algorithm to simulate nonlinear classical dynamics on a quantum computer. In particular, the manuscript describes an algorithm that implements the time evolution of the Lorenz model. This algorithm can reproduce both regular attractors and chaotic attractors within the chosen parameter regime.

The topic of the manuscript is interesting and has potential implications for quantum technology. The results presented in the submitted paper are interesting and valid enough to be published, and the authors provide a sufficient list of publications introducing the reader to the topics discussed. However, in my opinion, the current state of the manuscript is not suitable for publication.

1. In the reference section, I found eight papers by the authors of the manuscript. I think the number of self-citations is excessive. I recommend that the authors limit the number of self-citations.

2. The authors should add an explanation of the beta parameter below equation (3).

3. In lines 425-428, the authors write: "Implementing the quantum solver for the second-order finite difference scheme, we find excellent qualitative agreement between the 2nd-order generated chaotic attractor in Figure 12 and that generated by the adaptive grid higher order ODE solver used in Figure 11." Estimating visually, the two figures appear quite similar. The question is whether they are identical. Therefore, the authors should discuss in more detail how compatible the results obtained using their algorithm are with those obtained using the adaptive grid higher-order ODE solver.
The same question applies to Figures 13–16. To what extent is the method used compatible with other calculation methods?

4. In the conclusion, the authors should address whether the proposed algorithm has any limitations.

Reviewer 3 Report

Comments and Suggestions for Authors I have read ``A time-marching quantum algorithm for simulation of the nonlinear Lorenz dynamics'' by Kououtsis et al., manuscript entropy-3752824, with interest; but I must admit that while the technical derivations in the paper seem generally correct, I have significant concerns about the motivation for this problem and about the practicality of the solution.   The goal of this work is to develop a quantum algorithm to simulate the evolution of the Lorenz system (the famous 3D classical model that exhibits chaos and strange attractors) using the amplitudes of a quantum state to represent the values of the classical variables, and a quantum circuit to produce the system's time evolution. This is a challenging task, because the Lorenz system's equations of motion are nonlinear, whereas quantum systems evolve linearly, by unitary transformations. To get around this difficulty, the authors make use of a number of different ideas:   A. They first replace the continuous ODE with a discretized version, considering both first-order Euler and second-order Runge-Kutta integration; this is essentially the same approach used in classical simulation, and could be extended to higher-order algorithms in principle.   B. To obtain nonlinear evolution of a quantum state, they use a combination of two techniques: first, the Hadamard product of multiple copies of the state, and second, the ``linear combination of unitaries'' trick that uses a controlled unitary followed by a basis change and a measurement. An important note: *both* of these tricks work only probabilistically. To succeed, one must make a measurement of an extra register and postselect on obtaining a particular outcome (in this case, measuring a quantum register in the all-0 state).   C. A and B above are used to evolve the state for a single timestep. To evolve for multiple steps (as would be necessary for a long simulation) one must iterate this procedure. However, since multiple copies of the state are needed (three copies for the Lorenz system), iterating this procedure would quickly require an exponentially large number of copies of the initial state. To get around this, the authors use a recursive procedure: preparing three copies of the initial state, they make one copy at the next step, then reuse the qubits of the extra copies again to make another copy of the evolved state, and so forth until they have three copies of the evolved state, which they use to create one copy of the state at the next timestep; that copy is stored while they go through the procedure again, and so on. To evolve for N steps requires a number of qubits that grows linearly in N (though the *time* to execute the algorithm grows exponentially).   While I did not check every bit of the circuits and derivations, the approach seems like it will lead to the right answer. However, I have some serious concerns about both the motivation for this approach and for the practicality of this algorithm. Here they are, briefly:   1. Why do we *want* to solve the Lorenz system (or any such low-dimensional chaotic system) using quantum states? It is true that the size of the quantum system used, in qubits, is much smaller than the digital representation of the classical variables, in bits. But there are also real disadvantages to using a quantum state for a classical simulation. The state is not an observable, so we cannot read out the amplitudes that store the classical variables. The best one can do is run the simulation many times, measuring after each run, and collect statistics to estimate their values. Since the time complexity of this algorithm is already extremely high (see below) this makes the time required even worse.   While it is conceivable that a simulation like this could be used as part of another algorithm, with the quantum state provided as an input (rather than being measured to extract the classical information), no such application was suggested in the paper.   2. I have concerns about using a quantum state to represent any classical nonlinear system, especially a chaotic one. If we look at the representation in Eq. (20) of the current paper, the amplitudes of three basis states (assumed to be a subspace of a 3-qubit system) are used to represent the classical values x, y and z, with an overall normalization 1/sqrt(x^2 + y^2 + z^2). Clearly, any multiple of the vector, r(x,y,z), will be represented by the same state for any positive real number r. But in a nonlinear system, these different multiples do not evolve in the same way. Which one is the state representing?   3. The algorithm has a time complexity that grows exponentially in the number of time steps. This is not surprising, since to reach a given time step one must prepare three copies of the state at the previous time step, each of which required us to prepare three copies of the state at the time step before that, and so on. Exponential time complexity is the hallmark of an inefficient algorithm.   4. By contrast, the Lorenz system is *not* difficult to simulate classically. It is true that to match a particular solution for a given length of time requires that the initial state be represented with greater and greater precision as the time gets longer. But to study the long-time behavior of the system, such as the formation of the strange attractor, does not require that kind of precision, since it is the same (on a large scale) for any initial condition. The time complexity of classical simulation is essentially linear. (Simulating conservative systems is actually more difficult in a way than simulating a dissipative system like the Lorenz system, since discretization often violates the conservation law.)   5. In addition to the time-complexity being exponential, this algorithm is probabilistic: it succeeds if a final measurement gives the desired result. For most of the paper the authors do not estimate this success probability, but they give a formula for it in Eq. (67). For N time steps, the success probability scales like   (1/sigma^2)^(3^N) ,   where sigma^2 is a number slightly larger than 1 (plotted in Figure 9). Let's suppose that sigma^2 = 1 + Delta. Then this scaling goes like   exp(- Delta*3^N).   In other words, the probability of this successful postselection drops *doubly exponentially* with the number of integration steps. That scaling makes sense; the time complexity grows exponentially with N, and each of those steps has at least a small probability of failure. If Delta is small, this may not be too bad for a limited numbers of steps, but it will quickly doom this approach.   Because of these issues, I have serious concerns about publishing this paper. The technical result is interesting, and the way of combining the main ideas is clever, but the motivation is not obvious to me, and the algorithm seems to have major problems that would make it impractical for any real simulation. It is possible that there are ways around the concerns I have raised above, but they are not obvious.   In addition to the major issues above, I noted a few minor things as I went along:   While minor English usage issues are not uncommon, this paper also shows signs that it has not been carefully proofread or revised, such as repeated words (e.g., "previously established previously"). It needs to be thoroughly reread and revised by the authors.   P2, "These artifacts are difficult to control or even to eliminate them, hence comprising the accuracy of the simulation" I think you probably mean "These artifacts are difficult to eliminate or even to control, hence compromising the accuracy of the simulation"   What do the authors mean by "quantum-speeds"? Do they mean "quantum speed-ups," or something else? Comments on the Quality of English Language

This paper has a lot of minor grammatical errors, typos, and badly-phrased statements.

Round 2

Reviewer 2 Report

Comments and Suggestions for Authors

The authors have addressed all the concerns previously highlighted in my first review and the manuscript has been revised accordingly.
Therefore, I recommend the manuscript for publication.

Reviewer 3 Report

Comments and Suggestions for Authors

I've read the reply by the authors of ``A time-marching quantum algorithm for simulation of the nonlinear Lorenz dynamics,'' manuscript entropy-3752824, to my report on the previous draft of this paper, and taken a look at the revisions they have made to the manuscript in response. I appreciate that the authors acknowledge the points I have made, and that they are trying to work around them. Unfortunately, however, their answers don't seem sufficient to overcome problems of this magnitude. I will address these points, which I will number in the same way as in my first report.

1, 3 and 4. The authors have expanded on the motivation for this approach in their revised introduction. I note in particular a couple of sentences: they want to address ``whether quantum computers can be harnessed to speed-up the simulation of complex, nonlinear dissipative classical systems exhibiting classical turbulence and chaos,'' and regarding the algorithm presented in this paper, a ``key feature of the proposed quantum solver is the adoption of a quantum re-usage of states [19] resulting into a recursive structured quantum time-marching algorithm that maintains the previously established quantum speed-up while requiring only a linear number of state copies with respect to the total number of integration steps.'' But what quantum speed-up are they talking about? The Lorenz system can be numerically solved by a classical computer efficiently, up to the inevitable growth of imprecision accompanying any numerical solution of a chaotic system (which applies equally to the algorithm presented in this paper). The simulation time grows as a polynomial in the evolution time. By contrast, the quantum algorithm takes a time that grows exponentially in the evolution time--and that is *before* taking into account the rapidly diminishing success probability.

I am quite sympathetic to the possibility of exploring an algorithm that may not offer a quantum speed-up, in order to understand the problem better or develop new methods. But I don't see what the path is from what the authors are doing here to an algorithm that would offer a meaningful quantum advantage. It looks like a dead end. If it isn't, I would like to understand why not.

In fact, quantum computers can run the *classical* simulation algorithm perfectly well, in essentially the same way classical computers do: rather than map the values of the variables onto the amplitudes of a quantum state, one can use a digital approximation, mapping the variables to bit strings. There probably isn't a quantum advantage that way, but at least it isn't much worse than the classical algorithm.

2. The authors acknowledge that representing the 3-vector of the Lorenz system by the amplitudes of a quantum state as in Eq. (20) is many-to-one because of the state normalization, so the vectors (x,y,z) and (rx,ry,rz) would be mapped to the same state. They point out that the solution produced by their algorithm will only be correct for a single value of r, which I'd imagine is true. But that wasn't exactly the point of my comment: (x,y,z) and (rx,ry,rz) are both valid states of the Lorenz system. It seems that the algorithm can simulate the evolution of one of them, but not the other. That is a major limitation.

Moreover, the norm of the Lorenz vector (x,y,z) is not preserved by the evolution:

(d/dt)(x^2+y^2+z^2) = 2((sig+rho) xy - sig x^2 - y^2 - beta z^2) != 0.

So if the algorithm is working correctly then the value of r that maps between the values of the variables and the values of the amplitudes will change with time.

One could get around this, potentially, by embedding the system in a 4D Hilbert space, with a fixed r and the extra degree of freedom making the state always normalized. This would somewhat increase the number of runs needed to estimate the amplitudes, but that is a minor issue compared with the problems in point 5 below.

5. The success probability of one run of the algorithm decays *doubly* exponentially. There is not an easy workaround for this problem. I have calculated that if 1/sigma^2 = 0.9999--better than it is likely to be in practice--then after 12 steps the success probability is of the order 10^-23. That is for 12 steps, when the timestep Delta t is small. The authors suggest that amplitude amplification could be used to magnify the success probability. But the complexity of amplitude amplification is sqrt(T); that is, to amplify the success probability close to 1, the simulation circuit for 12 steps would have to be repeated more than 10^11 times. And increasing the number of steps by 1, to 13 from 12, basically cubes the size of the problem, so it would require more than 10^34 repetitions of the circuit. It's simply not possible to overcome a problem like this using amplitude amplification.

These are basically the same concerns I raised in my first report; I don't think that minor changes to the algorithm can fix them. For these reasons, I cannot recommend publication of this article. The authors need to figure out something with much higher success probability, at a minimum, before this research could be of interest.

Comments on the Quality of English Language

Some of the grammatical errors, etc., have been fixed, but overall some improvement is still possible.